# The RAB27A effector SYTL5 regulates mitophagy and mitochondrial metabolism

Ana Lapão[1,2†], Lauren Sophie Johnson[1,2,3†], Laura Trachsel-Moncho[1,2,3], Samuel J Rodgers[2,3], Sakshi Singh[1,2], Matthew YW Ng[1,2], Sigve Nakken[2,4,5], Eeva-Liisa Eskelinen[6], Anne Simonsen[1,2,3*]

[1]Department of Molecular Medicine, Institute of Basic Medical Sciences, Faculty of Medicine, University of Oslo, Oslo, Norway; [2]Centre for Cancer Cell Reprogramming, Institute of Clinical Medicine, Faculty of Medicine, University of Oslo, Oslo, Norway; [3]Department of Molecular Cell Biology, Institute for Cancer Research, Oslo University Hospital Montebello, Oslo, Norway; [4]Department of Tumour Biology, Institute for Cancer Research, Oslo University Hospital, Oslo, Norway; [5]Department of Informatics, University of Oslo, Oslo, Norway; [6]Institute of Biomedicine, University of Turku, Turku, Finland

## eLife Assessment

This study by Lapao et al. uncovers a novel role for the Rab27A effector SYTL5 in regulating mitochondrial function and mitophagy under hypoxic conditions. Using a range of imaging and functional assays, the authors demonstrate that SYTL5 localizes to mitochondria in a Rab27A-dependent manner and impacts mitochondrial respiration and metabolic reprogramming. While the findings are **solid** and **valuable** in the area of cancer biology, further mechanistic clarity and improved imaging would strengthen the conclusions.

**\*For correspondence:**
anne.simonsen@medisin.uio.no

[†]These authors contributed equally to this work

**Abstract** SYTL5 is a member of the Synaptotagmin-Like (SYTL) protein family that differs from the Synaptotagmin family by having a unique N-terminal Synaptotagmin homology domain that directly interacts with the small GTPase RAB27A. Several SYTL protein family members have been implicated in plasma membrane transport and exocytosis, but the specific function of SYTL5 remains unknown. We here show that SYTL5 is a RAB27A effector and that both proteins localise to mitochondria and vesicles containing mitochondrial material. Mitochondrial recruitment of SYTL5 depends on its interaction with functional RAB27A. We demonstrate that SYTL5-RAB27A positive vesicles containing mitochondrial material, autophagy proteins and LAMP1 form during hypoxia and that depletion of SYTL5 and RAB27A reduces mitophagy under hypoxia mimicking conditions, indicating a role for these proteins in mitophagy. Indeed, we find that SYTL5 interacts with proteins involved in vesicle-mediated transport and cellular response to stress and that its depletion compromises mitochondrial respiration and increases glucose uptake. Intriguingly, SYTL5 expression is significantly reduced in tumours of the adrenal gland and correlates positively with survival for patients with adrenocortical carcinoma.

## Introduction

The Synaptotagmin-Like (SYTL) protein family consists of 5 members (SYTL1-5), each containing an N-terminal Synaptotagmin homology domain (SHD) and two C-terminal C2 lipid-binding domains. The SHD domains of SYTL1-5 have been found to directly interact with the small GTPase protein RAB27 (*Kuroda et al., 2002a*; *Kuroda et al., 2002b*). The C2 domain is generally involved in

phospholipid binding and interacts with cellular membranes (*Nalefski and Falke, 1996*), either in a calcium-dependent or calcium-independent manner (*Nalefski and Falke, 1996*; *Gálvez-Santisteban et al., 2012*). Calcium was shown to be required for SYTL3 and SYTL5 phospholipid binding activity (*Kuroda et al., 2002a*; *Fukuda, 2002*), whereas SYTL2 binding to phosphatidylserine (PS) was inhibited by calcium (*Yu et al., 2007*).

The SYTL protein family is generally involved in plasma membrane transport and exocytosis (*Kuroda et al., 2002b*; *Gálvez-Santisteban et al., 2012*), mostly through their binding to RAB27 proteins. The main functions of RAB27 are related to vesicle budding, delivery, tethering and fusion with membranes (*Fukuda, 2013*), and its activity is regulated by a cyclic activation and inactivation state depending on its binding to guanosine-5'-triphosphate (GTP) or guanosine diphosphate (GDP), respectively (*Fukuda, 2013*). RAB27A and RAB27B are the two RAB27 isoforms found in vertebrates (*Fukuda, 2013*), and their binding to effector proteins occurs only when they are bound to GTP (*Fukuda, 2013*). SYTL5 binds to the GTP-bound RAB27A form, indicating its possible role in membrane trafficking events (*Kuroda et al., 2002a*).

Functional mitochondria are fundamental for normal cellular metabolism. Mitochondria are the primary source of adenosine triphosphate (ATP) obtained through oxidative phosphorylation (OXPHOS) and are important for cellular calcium homeostasis, lipid metabolism, reactive oxygen species (ROS) generation, and detoxification (*Montava-Garriga and Ganley, 2020*). Mitochondria can integrate and generate signalling cues to adjust their metabolism and biogenesis to maintain cellular homeostasis (*Montava-Garriga and Ganley, 2020*; *Martínez-Reyes and Chandel, 2020*). Mitochondrial function and health are tightly regulated by several quality control processes involving targeting of parts of mitochondria to lysosomes for degradation, including macromitophagy (*Allen et al., 2013*; *Lazarou et al., 2015*; selective degradation of mitochondria by autophagy), piecemeal mitophagy (*Le Guerroué et al., 2017*; *Abudu et al., 2021*), mitochondrial-derived vesicles (MDVs; *Neuspiel et al., 2008*; *Soubannier et al., 2012*), and vesicles derived from the inner mitochondrial membrane (VDIMs; *Prashar et al., 2024*). It has also been found that damaged mitochondria can be directly released to the extracellular space (*Choong et al., 2021*), either contained within extracellular vesicles (*Phinney et al., 2015*) or by mitocytosis, where damaged mitochondria are expelled in migrasomes (*Jiao et al., 2021*).

Exposure of mitochondria to various stressors can affect their function and lead to severe diseases, such as neurodegenerative disorders and cancer (*Youle and Narendra, 2011*; *Ding and Yin, 2012*). Cancer cells can trigger a change in mitochondrial metabolism to promote cell proliferation and survival by a process known as the Warburg effect, characterised by a switch from OXPHOS to ATP production through glycolysis even under normoxic conditions (*Liberti and Locasale, 2016*). Adrenocortical carcinoma (ACC) is a rare type of cancer that develops in the adrenal gland cortex, having a very poor prognosis and limited treatment options (*Else et al., 2014*). Most patients are diagnosed at advanced cancer stages and present an excess in adrenocortical hormone production (*Else et al., 2014*). Mitochondria are known to be essential organelles in the synthesis of steroid hormones, including cortisol, since they contain specific enzymes that catalyse steroid synthesis from cholesterol delivered to the mitochondria (*Miller, 2013*).

Here, we demonstrate that SYTL5 localises to mitochondria in a RAB27A-dependent manner and that both proteins regulate mitophagy and cellular metabolism. Cells lacking SYTL5 undergo a shift from mitochondrial oxygen consumption to glycolysis, which may explain the correlation between low SYTL5 expression and poor survival of ACC patients.

## Results

### SYTL5 localises to mitochondria and endolysosomal compartments

SYTL5 was identified as a putative candidate in a screen for lipid-binding proteins involved in the regulation of mitophagy (*Munson et al., 2021*). To characterise the cellular localisation and function of SYTL5, we generated U2OS cells with stable inducible expression of SYTL5-EGFP, as there are no antibodies recognising endogenous SYTL5 and our efforts to tag endogenous SYTL5 using CRISPR/Cas9 were also unsuccessful. Intriguingly, live cell imaging analysis revealed that SYTL5-EGFP co-localised with filamentous structures positive for MitoTracker red (*Figure 1A*), which labels active mitochondria. Moreover, several SYTL5-EGFP-positive vesicles were observed, including some small

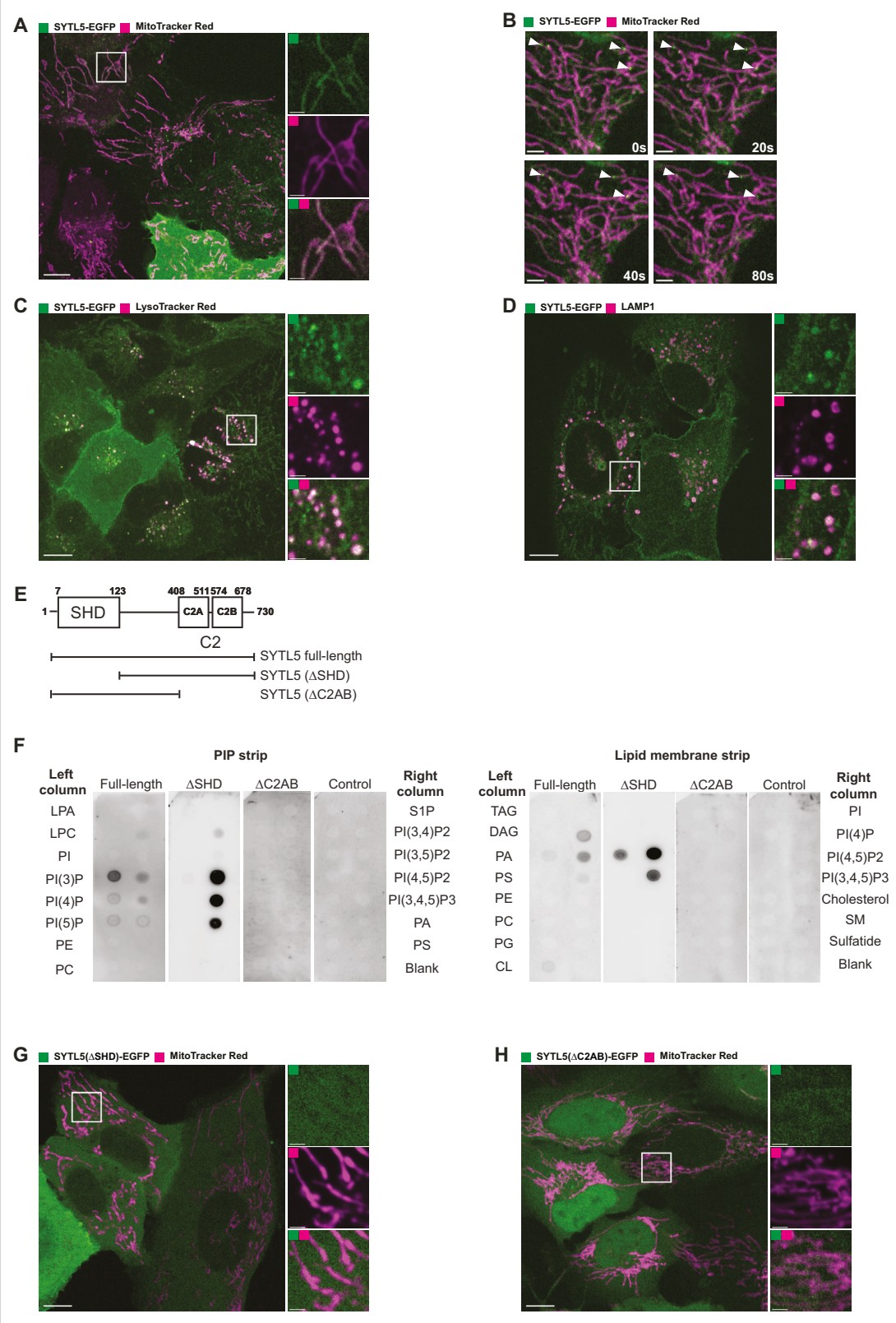

**Figure 1.** Lipid binding protein SYTL5 localises to mitochondria and endo-lysosomes. (**A**) Live confocal microscopy imaging of U2OS stably expressing SYTL5-EGFP co-stained with 50 nM MitoTracker red. SYTL5-EGFP expression was induced for 24 hr using 100 ng/ml doxycycline. MitoTracker Red was added 30 min before imaging. Scale bars: 10 µm, 2 µm (insets). (**B**) Time-lapse video frames (*Figure 1—video 1*) tracking SYTL5 vesicle movement along filaments positive for MitoTracker red (arrows). SYTL5-EGFP expression was induced for 24 hr using 100 ng/ml doxycycline. Scale bars: 3 µm.

*Figure 1 continued on next page*

*Figure 1 continued*

(**C**) Live confocal microscopy imaging of U2OS stably expressing SYTL5-EGFP co-stained with 50 nM LysoTracker red. SYTL5-EGFP expression was induced for 24 hr using 100 ng/ml doxycycline. LysoTracker Red was added 30 min before imaging. Scale bars: 10 μm, 2 μm (insets). (**D**) Confocal imaging of U2OS stably expressing SYTL5-EGFP stained for endogenous LAMP1. Nuclei were stained using Hoechst. Scale bars: 10 μm, 2 μm (insets). (**E**) Overview of domain structure of SYTL5 full length, ΔSHD and ΔC2AB mutants. (**F**) 3xFLAG fusion proteins expressed in U2OS cells (SYTL5-EGFP-3xFLAG, SYTL5 (ΔSHD)-EGFP-3xFLAG and SYTL5 (ΔC2AB)-EGFP-3xFLAG) were immunoprecipitated and added to membranes spotted with lipids found in cell membranes: LPA (lipoprotein A); LPC (lysophosphatidylcholine); PI (phosphatidylinositol); PI(3)P (phosphatidylinositol 3-phosphate); PI(4)P (phosphatidylinositol 4-phosphate); PI(5)P (phosphatidylinositol 5-phosphate); PE (phosphatidylethanolamine); PC (phosphatidylcholine); S1P (sphingosine-1-phosphate); PI(3,4)P2 (phosphatidylinositol 3,4-bisphosphate); PI(3,5)P2 (phosphatidylinositol 3,5-bisphosphate); PI(4,5)P2 (phosphatidylinositol 4,5-bisphosphate); PI(3,4,5)P3 (phosphatidylinositol 3,4,5-triphosphate); PA (phosphatidic acid); PS (phosphatidylserine); TAG (triglyceride); DAG (diglyceride); PG (phosphatidylglycerol); CL (cardiolipin); Cholesterol; SM (sphingomyelin); Sulfatide. (**G**) Live confocal microscopy imaging of U2OS stably expressing SYTL5 (ΔSHD)-EGFP-3xFLAG co-stained with 50 nM MitoTracker red added 30 min before imaging. Scale bars: 10 μm, 2 μm (insets). (**H**) Live confocal microscopy imaging of U2OS stably expressing SYTL5 (ΔC2AB)-EGFP-3xFLAG co-stained with 50 nM MitoTracker red added 30 min before imaging. Scale bars: 10 μm, 2 μm (insets).

The online version of this article includes the following video, source data, and figure supplement(s) for figure 1:

**Source data 1.** Original image files for *Figure 1A, C, D, G and H*.

**Source data 2.** Original uncropped membranes for *Figure 1F*.

**Source data 3.** Uncropped membranes with the relevant spots clearly labelled for *Figure 1F*.

**Figure supplement 1.** Co-localisation of SYTL5 with RAB proteins.

**Figure supplement 1—source data 1.** Original image files for *Figure 1—figure supplement 1A, B, C, D, E, F*.

**Figure 1—video 1.** Live confocal microscopy imaging tracking SYTL5 vesicle movement along filaments positive for MitoTracker red.

https://elifesciences.org/articles/105541/figures#fig1video1

and highly mobile SYTL5-EGFP vesicles moving along the mitochondrial network (*Figure 1—video 1* and arrows in *Figure 1B*). SYTL5-EGFP staining at the plasma membrane was also seen, particularly in cells with higher expression levels (*Figure 1C*), which may suggest a role for SYTL5 in secretion to the plasma membrane.

To determine the identity of the SYTL5-positive vesicles, U2OS SYTL5-EGFP cells were infected with a panel of lentiviral RAB GTPase constructs representing different cellular compartments (*Figure 1—figure supplement 1A–F*) and analysed by live cell imaging. SYTL5-EGFP-positive structures were observed to co-localise with vesicles positive for RAB4 and RAB11, two GTPases that mainly localise to recycling endosomes (*Wandinger-Ness and Zerial, 2014*) and to some extent with RAB5, mostly localised to early endosomes (*Wandinger-Ness and Zerial, 2014*; *Figure 1—figure supplement 1A–C*). Co-localisation was also observed with RAB7, a marker of late endosomes, lysosomes, and autophagosomes (*Wandinger-Ness and Zerial, 2014*) and partially with RAB9, which associates with late endosomes and mediates the transport between late endosomes and the *trans*-Golgi network (*Wandinger-Ness and Zerial, 2014*; *Figure 1—figure supplement 1D–E*). SYTL5-EGFP showed little or no co-localisation with RAB6-positive structures, representing Golgi-derived vesicles (*Figure 1—figure supplement 1F*). In line with an endocytic identity of SYTL5-EGFP-positive vesicles, they were also positive for lysosomal markers, including LysoTracker red, which labels acidic compartments in the cell (*Figure 1C*), and the lysosomal membrane marker LAMP1 (*Figure 1D*).

Thus, based on live cell imaging analysis, we conclude that SYTL5 localises to the mitochondrial network and to mitochondria-associated vesicles that partly overlap with endolysosomal compartments.

## Mitochondrial localisation of SYTL5 requires both the RAB27-binding SHD domain and the lipid-binding C2 domains

To investigate whether the observed intracellular localisation of SYTL5-EGFP depends on its binding to RAB27 and/or lipids, we generated U2OS cells with stable expression of EGFP-3xFLAG-tagged wild type SYTL5 or SYTL5 lacking either the RAB27-binding SHD domain (SYTL5(ΔSHD)) or the two lipid-binding C2 domains (SYTL5(ΔC2AB); *Figure 1E*).

To validate the SYTL5(ΔC2AB) mutant and determine the lipid-binding specificity of SYTL5, lysates from these cells were incubated with membranes containing various phosphoinositides and other lipids. Full-length SYTL5 was observed to bind to mono-phosphorylated phosphoinositides (PI(3)P,

PI(4)P, PI(5)P), as well as PI(4,5)P2, PI(3,4,5)P3 and to some extent to phosphatidic acid (PA) and cardiolipin (CL) (*Figure 1F*). As expected, upon removal of the C2 domains, specific binding to all lipid species was lost, while SYTL5 lacking the SHD domain retained the ability to bind to lipids (*Figure 1F*).

The SYTL5(ΔSHD)-EGFP and SYTL5(ΔC2AB)-EGFP expressing cell lines were then incubated with MitoTracker red and analysed by live cell imaging. Indeed, the SYTL5 colocalisation to mitochondria was lost, and both SYTL5(ΔSHD)-EGFP and SYTL5(ΔC2AB)-EGFP were dispersed in the cytosol (*Figure 1G–H*), indicating that both the RAB27-interacting SHD domain and the lipid binding C2 domains are required for mitochondrial localisation of SYTL5.

## Mitochondrial localisation of SYTL5 requires RAB27A GTPase activity

Given that the SHD domain of SYTL5 has been shown to bind RAB27A (*Kuroda et al., 2002a*) and our observations that the SHD domain is required for mitochondrial localisation of SYTL5, we asked whether RAB27A activity is required for SYTL5 recruitment to mitochondria. U2OS SYTL5-EGFP cells were infected with a mScarlet-RAB27A lentiviral construct to generate a stable cell line. Co-expression of SYTL5-EGFP and mScarlet-RAB27A resulted in a clear co-localisation of both proteins to filaments positive for MitoTracker DeepRed (DR) (*Figure 2A*). The mitochondrial localisation of SYTL5-EGFP and mScarlet-RAB27A was confirmed by correlative light and EM (CLEM) analysis using the same cell line (*Figure 2B*, zoom in 1). Intriguingly, the mitochondrial localisation of SYTL5-EGFP was enhanced when co-expressed with mScarlet-RAB27A compared to cells expressing SYTL5-EGFP only (*Figure 1A–B*), indicating that RAB27A might facilitate mitochondrial recruitment of SYTL5. Indeed, in U2OS cells with stable expression of mScarlet-RAB27A only, RAB27A strongly co-localised with MitoTracker green (*Figure 2—figure supplement 1A*), demonstrating its mitochondrial targeting. Moreover, SYTL5-EGFP and mScarlet-RAB27A were both detected in the mitochondrial fraction (containing TIM23 and COXIV) of cells expressing SYTL5-EGFP or mScarlet-RAB27A (*Figure 2—figure supplement 1B*). Importantly, the mitochondrial localisation of SYTL5 and RAB27A is not specific to U2OS cells as HeLa cells with stable expression of SYTL5-EGFP or mScarlet-RAB27A show a similar expression pattern to U2OS cells, as demonstrated by co-localisation of SYTL5-EGFP with MitoTracker Red and plasma membrane staining in cells having higher expression (*Figure 2—figure supplement 1C*) and of mScarlet-RAB27A with MitoTracker Green (*Figure 2—figure supplement 1D*). All further experiments were conducted in U2OS cells.

To further characterise mitochondrial recruitment of SYTL5 and RAB27A and elucidate their role at the mitochondrion, we used CRISPR/Cas9 to knock out (KO) RAB27A and SYTL5 in U2OS cells (*Figure 2—figure supplement 2A–D*). This double-KO (dKO) cell line was further transduced with lentiviral constructs to constitutively express SYTL5-EGFP, mScarlet-RAB27A, or both, and their mitochondrial localisation was analysed by live cell imaging (*Figure 2C–E*). Intriguingly, mScarlet-RAB27A co-localised extensively with MitoTracker Green and to small vesicles dispersed in the cytoplasm and located at or near mitochondria (*Figure 2C*), indicating that SYTL5 is not required for RAB27A mitochondrial localisation. In contrast, SYTL5-EGFP was not recruited to mitochondria when expressed alone in dKO cells, although some SYTL5-EGFP vesicles were seen near mitochondrial structures (*Figure 2D*). Upon co-expression of SYTL5-EGFP and mScarlet-RAB27A in dKO cells, the mitochondrial localisation of SYTL5-EGFP was rescued (*Figure 2E*), demonstrating that mitochondrial recruitment of SYTL5 is dependent on RAB27A.

Since SYTL5 has previously been described as a RAB27A effector (*Kuroda et al., 2002a*), we asked whether its mitochondrial recruitment depends on binding to the GTP-bound form of RAB27A. To address this, dKO cells expressing SYTL5-EGFP were rescued with the constitutively active (RAB27A-Q78L) or inactive (RAB27A-T23N) mutant forms of mScarlet-RAB27A. As expected, SYTL5-EGFP interacted specifically with RAB27A wild type (WT) and RAB27A-Q78L, as well as with endogenous RAB27A in control cells, while no interaction was detected with RAB27A-T23N, as assessed by GFP-pulldown (*Figure 2F*). To our surprise, live cell imaging of the same cells revealed that, in contrast to mScarlet-RAB27A WT, neither the RAB27A Q78L nor the T23N mutant localised to the mitochondrial network when expressed together with SYTL5-EGFP in dKO cells (*Figure 2E*). Also, SYTL5-EGFP failed to localise to mitochondria in cells expressing RAB27A Q78L, further demonstrating that mitochondrial recruitment of SYTL5 depends on mitochondrial RAB27A localisation (*Figure 2E*). Taken together, our data indicate that the GTPase activity of RAB27A is required for its mitochondrial localisation, which facilitates further recruitment of SYTL5.

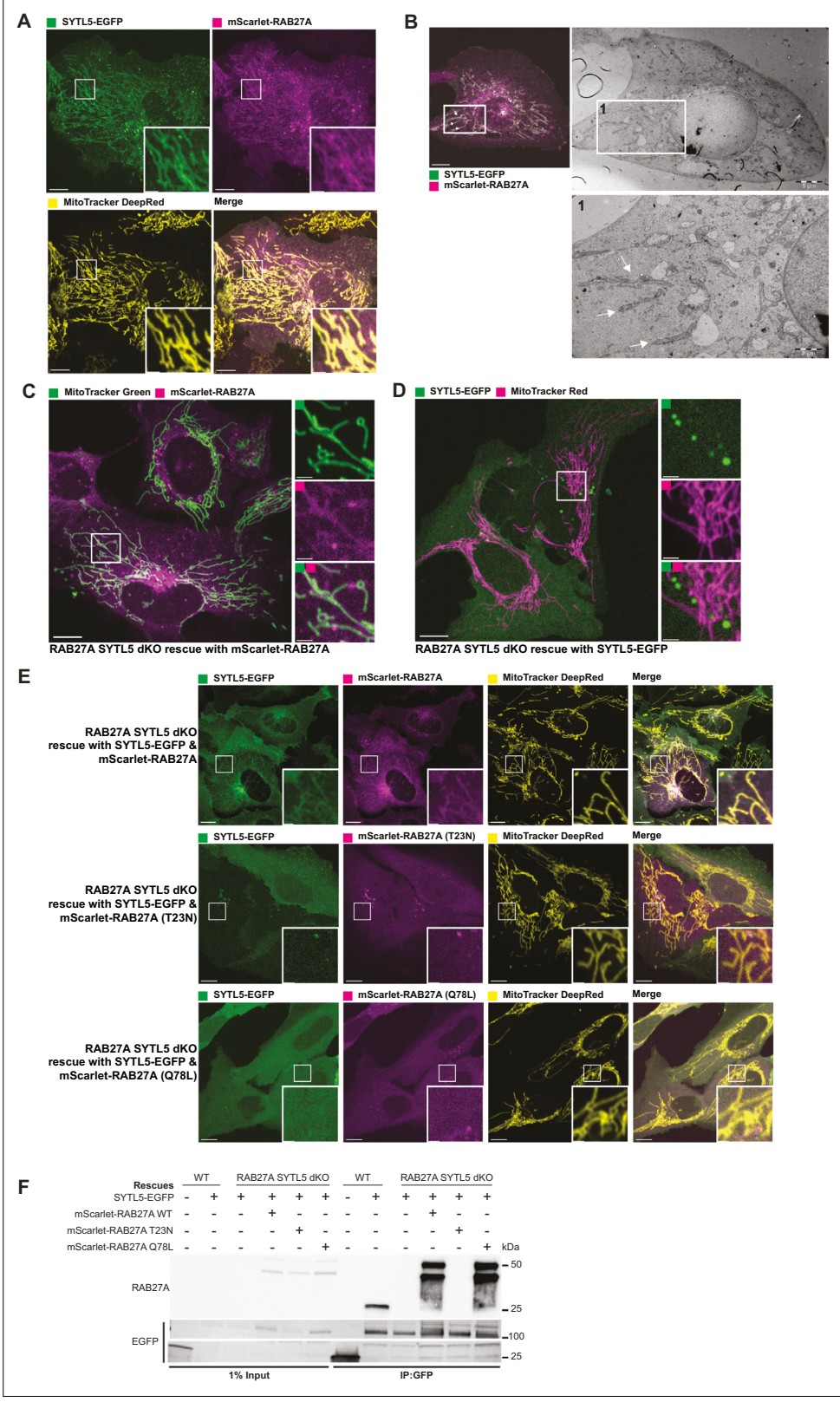

**Figure 2.** Mitochondrial localisation of SYTL5 requires RAB27A GTPase activity. (**A**) SYTL5-EGFP expression was induced for 24 hr with 100 ng/ml doxycycline in U2OS cells with stable inducible expression of SYTL5-EGFP and constitutive expression of mScarlet-RAB27A. Cells were co-stained with MitoTracker DR for 30 min before live imaging. Arrows indicate mitochondrion filaments. Scale bars: 10 μm, 2 μm (insets). (**B**) CLEM analysis of the cells

*Figure 2 continued on next page*

*Figure 2 continued*

described in A. Before imaging, cells growing in monolayer were fixed in warm (≈37 °C) 3.7% paraformaldehyde in 0.2 M HEPES (pH 7). After fixation, cells were imaged using a confocal microscope to acquire Z-stacks of optical sections and DIC images to locate the cells of interest. The cells were finally fixed using 2% glutaraldehyde in 0.2 M HEPES (pH 7.4) for 120 min before sample preparation for TEM. Arrows in panel 1 indicate mitochondrion filaments. Scale bars: 15 µm (left), 5 µm (middle) and 2 µm (right). (C) U2OS dKO cells were rescued with mScarlet-RAB27A and co-stained with 50 nM MitoTracker green for 30 min before imaging. Scale bars: 10 µm, 2 µm (insets). (D) U2OS dKO cells were rescued with SYTL5-EGFP and co-stained with 50 nM MitoTracker red for 30 min before imaging. Scale bars: 10 µm, 2 µm (insets). (E) Live confocal microscopy imaging of U2OS dKO cells rescued with SYTL5-EGFP and mScarlet-RAB27A (upper panel); SYTL5-EGFP and mScarlet-RAB27A-T23N (middle panel) or SYTL5-EGFP and mScarlet-RAB27A-Q78L (lower panel). All cells were co-stained with 50 nM MitoTracker DR for 30 min before imaging. Scale bars: 10 µm, 2 µm (insets). (F) Lysates from U2OS cells stably expressing EGFP (control) and U2OS dKO cells expressing SYTL5-EGFP and/or mScarlet-RAB27A (wild-type, T23N, or Q78L mutants) were immunoprecipitated using GFP-Trap beads and analysed by western blot using RAB27A and EGFP antibodies.

The online version of this article includes the following source data and figure supplement(s) for figure 2:

**Source data 1.** Original image files for *Figure 2A, B, C, D and E*.

**Source data 2.** Original uncropped blots for *Figure 2F*.

**Source data 3.** Uncropped blots with the relevant bands clearly labelled for *Figure 2F*.

**Figure supplement 1.** Mitochondrial localisation of SYTL5 and RAB27A.

**Figure supplement 1—source data 1.** Original image files for *Figure 2—figure supplement 1A, C, D*, C, D.

**Figure supplement 1—source data 2.** Original uncropped blots for *Figure 2—figure supplement 1B*.

**Figure supplement 1—source data 3.** Uncropped blots with the relevant bands clearly labelled for *Figure 2— figure supplement 1B*.

**Figure supplement 2.** Generation of RAB27A and SYTL5 knock out cells.

**Figure supplement 2—source data 1.** Original uncropped blots and gel for *Figure 2—figure supplement 2B, D*.

**Figure supplement 2—source data 2.** Uncropped blots with the relevant bands clearly labelled for *Figure 2— figure supplement 2B, D*.

## SYTL5 interacts with proteins involved in vesicle-mediated transport and cellular response to stress

As few molecular interactors of SYTL5 are known (*Kuroda et al., 2002a*), we analysed the SYTL5 interactome by immunoprecipitating SYTL5-EGFP or an EGFP control, followed by mass spectrometry (MS) and protein-hit enrichment analysis. RAB27A was one of the most significant hits found, providing evidence that the IP was successful, together with several proteins known to participate in secretion such as PDCD6IP (*Baietti et al., 2012*; *Figure 3A*). In total, 163 proteins were identified as significant SYTL5-EGFP interactors compared to the EGFP control (*Figure 3A*, *Supplementary file 1*) and were subjected to Gene Ontology (GO) term enrichment analysis. Considering GO enrichment within the subcellular compartment ontology (GO-CC), we discovered the following compartments as enriched (*Figure 3B*) (number of hits for each category in parentheses): cytoskeleton (43), intracellular vesicle (38), extracellular vesicle (36), plasma-membrane-bound cell projection (33), endosome (18), and focal adhesion (17). Furthermore, for biological processes, encoded in the GO-BP subontology, we discovered the following categories as enriched (*Figure 3A and C*): vesicle-mediated transport (38), cellular response to stress (34), response to oxygen-containing compound (31), intracellular protein transport (22), secretion (21), autophagy (13), stress-activated MAPK cascade (10), cellular response to oxidative stress (9), reactive oxygen species metabolic process (7), regulation of mitochondrion organisation (6), endosome organisation (4), and protein insertion into mitochondrial membrane (3). Taken together, the SYTL5 interactome implies a role for SYTL5 in mitochondrial processes and cellular response to stress, in line with its localisation to mitochondria, as well as a possible role in secretion and endocytosis, in line with its localisation to endocytic compartments and the plasma membrane (*Figure 1*).

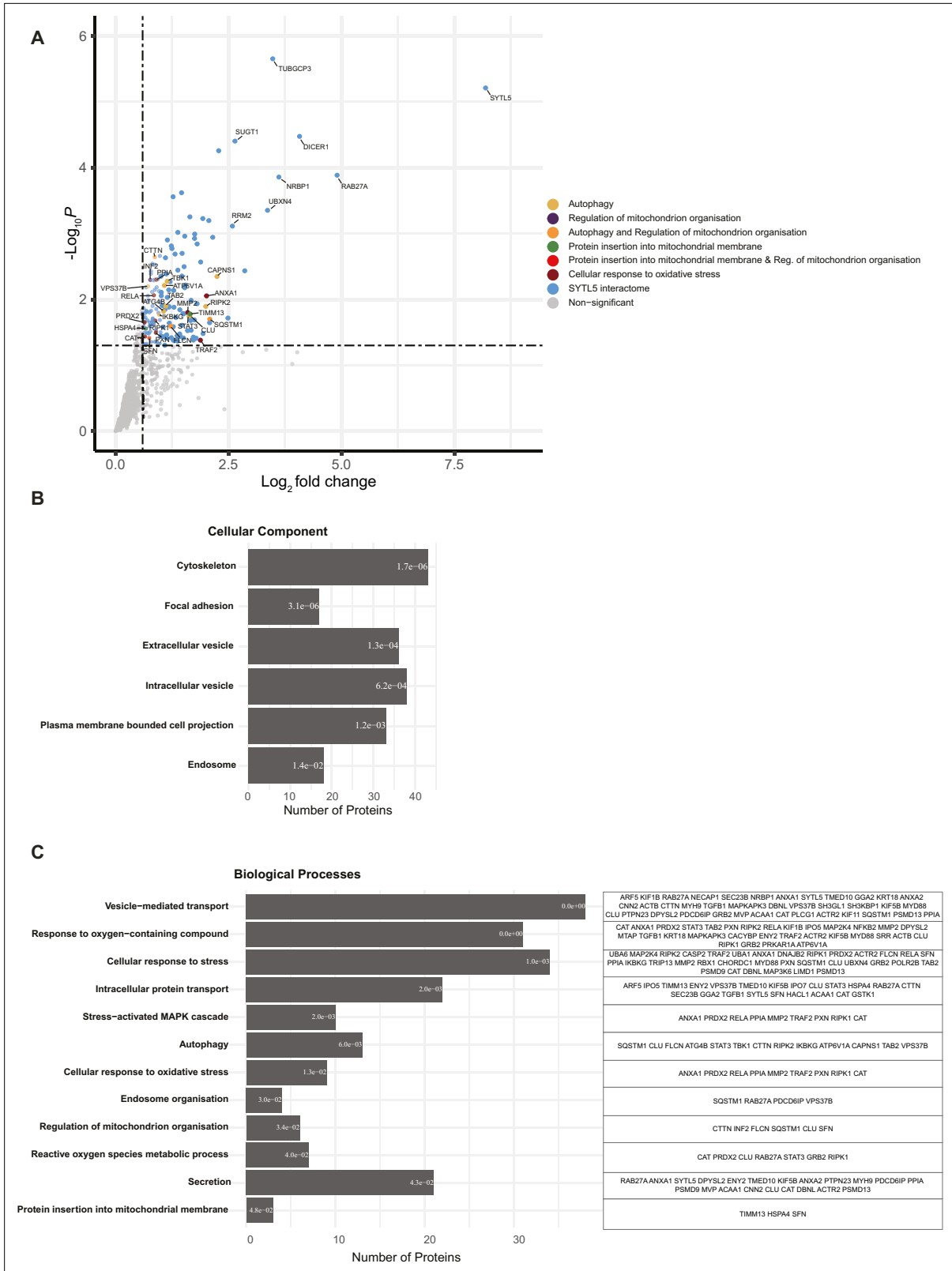

**Figure 3.** SYTL5 interacts with proteins involved in vesicle-mediated transport and cellular response to stress. (**A**) SYTL5 interactome. SYTL5-EGFP expression in U2OS cells was induced for 24 hr using 100 ng/ml doxycycline, followed by immunoprecipitation of SYTL5-EGFP using GFP-Trap beads and identification of co-purified proteins by MS analysis. The list of co-purified proteins was compared with that obtained from cells expressing EGFP, and only significant (p<0.05) SYTL5-EGFP-specific protein hits are highlighted in colour. Protein hits falling into at least one of the biological processes:

*Figure 3 continued on next page*

*Figure 3 continued*

autophagy, regulation of mitochondrion organisation, protein insertion into mitochondrial membrane, or cellular response to oxidative stress are colour coded according to plot legend. All other significant hits are indicated in blue. Non-significant (p>0.05) hits are indicated in grey. Data are from three biological replicates. (**B**) GO cellular compartment term enrichment of significant protein hits co-purified with SYTL5-EGFP. Corresponding enrichment false discovery rate (FDR) value is represented inside each bar and bars are ordered from smallest to largest FDR (q-value) from top to bottom. (**C**) GO biological processes term enrichment of significant protein hits co-purified with SYTL5-EGFP. Corresponding enrichment FDR value is represented inside each bar and bars are ordered from smallest to largest FDR (q-value) from top to bottom. Proteins corresponding to each biological process category are listed in the table to the right.

## SYTL5-RAB27A-positive vesicles contain mitochondrial components

Several proteins involved in autophagy were identified as specific SYTL5 interactors, including SQSTM1/p62, ATG4B, TBK1, and ATP6V1A (*Figure 3A*), and given the mitochondrial localisation of SYTL5, we first investigated a possible role of SYTL5 in mitophagy. To induce mitophagy, dKO cells expressing mScarlet-RAB27A and SYTL5-EGFP were subjected to hypoxia or the drugs DFP (deferiprone, an iron chelator) or DMOG (Dimethyloxalylglycine, a prolyl-4-hydroxylase inhibitor) that mimics hypoxia by activation of HIF1α. We noticed an increase in the number and size of vesicles positive for SYTL5-EGFP and mScarlet-RAB27A that also contained MitoTracker Deep Red (MitoTracker DR) in all conditions, compared to control cells (*Figure 4A*).

HIF-1α is a transcription factor that is a main driver of the Warburg effect (*Soubannier et al., 2012*) and an inducer of mitophagy via upregulation of the outer mitochondrial membrane proteins BNIP3 and BNIP3L, which bind to LC3. To identify the nature of the SYTL5/RAB27A/MitoTracker DR-positive vesicular structures that increase with HIF-1α stabilisation, dKO cells expressing mScarlet-RAB27A and SYTL5-EGFP were immunostained with antibodies against various autophagy markers. Intriguingly, while observing strong co-localisation of SYTL5/RAB27A/MitoTracker DR-positive vesicles with LAMP1 (*Figure 4B*), the same structures did not co-localise with LC3B or the autophagy receptor p62 (*Figure 4—figure supplement 1A*). However, as both LC3 and p62 are degraded in the lysosome, we added the lysosomal V-ATPase inhibitor Bafilomycin A1 (BafA1) for the last 4 hr of DFP treatment, resulting in the presence of some LC3 and p62 positive SYTL5/RAB27A/MitoTracker DR vesicles (*Figure 4—figure supplement 1B*). It should be noted that the SYTL5/RAB27A/MitoTracker-positive vesicle structures do not always colocalise with LAMP1, LC3, and p62, and that the phenotype of these cells is varied. The mitochondrial localisation of SYTL5 and RAB27A was not affected in cells treated with the ULK1 inhibitor MRT68291 or with the class III phosphoinositide 3-kinase VPS34 inhibitor IN1 (*Figure 4—figure supplement 1C*), indicating that SYTL5 and RAB27A are recruited to mitochondria independent of active autophagy. Thus, our data show that SYTL5 and RAB27A-positive vesicles containing mitochondrial material and autophagic proteins form upon HIF-1α-induced mitophagy.

## SYTL5 and RAB27A function as positive regulators of selective mitophagy

To investigate whether SYTL5 and RAB27A are required for the turnover of mitochondrial proteins in hypoxic conditions, we took advantage of two mitophagy reporter U2OS cell lines, one expressing the mitochondrial matrix reporter, pSu9-Halo-mGFP (*Yim et al., 2022*) and one expressing the mitochondrial targeting signal of the matrix protein NIPSNAP1 tagged with EGFP-mCherry (*Princely Abudu et al., 2019*; hereafter referred to as IMLS cells).

The pSu9-Halo-mGFP cells allow a measure of mitophagy flux via the production of a cleaved Halo Tag band, which is stable in lysosomes when bound to the ligand (Halo$^{TMR}$). Activation of HIF1α by treatment with DFP or DMOG resulted in the appearance of cleaved Halo$^{TMR}$ (*Figure 4C–D*), consistent with activation of mitophagy. The use of small interfering RNA (siRNA) to deplete SYTL5, RAB27A or both simultaneously in these cells resulted in a significant reduction in the amount of free Halo$^{TMR}$ (*Figure 4C–D*), highlighting a reduction in mitophagy. Depletion of SYTL5 and/or RAB27A did not affect the DFP or DMOG-induced HIF1α activation, as analysed by BNIP3L expression levels (*Figure 4—figure supplement 2A*). Thus, both SYTL5 and RAB27A are required for the lysosomal turnover of mitochondrial components in response to HIF1α activation and not for the HIF1α-mediated induction of mitophagy.

In contrast, we did not observe a significant difference in mitophagy levels in IMLS cells depleted of SYTL5 or RAB27A treated with DFP to induce Parkin-independent mitophagy, or in IMLS cells with

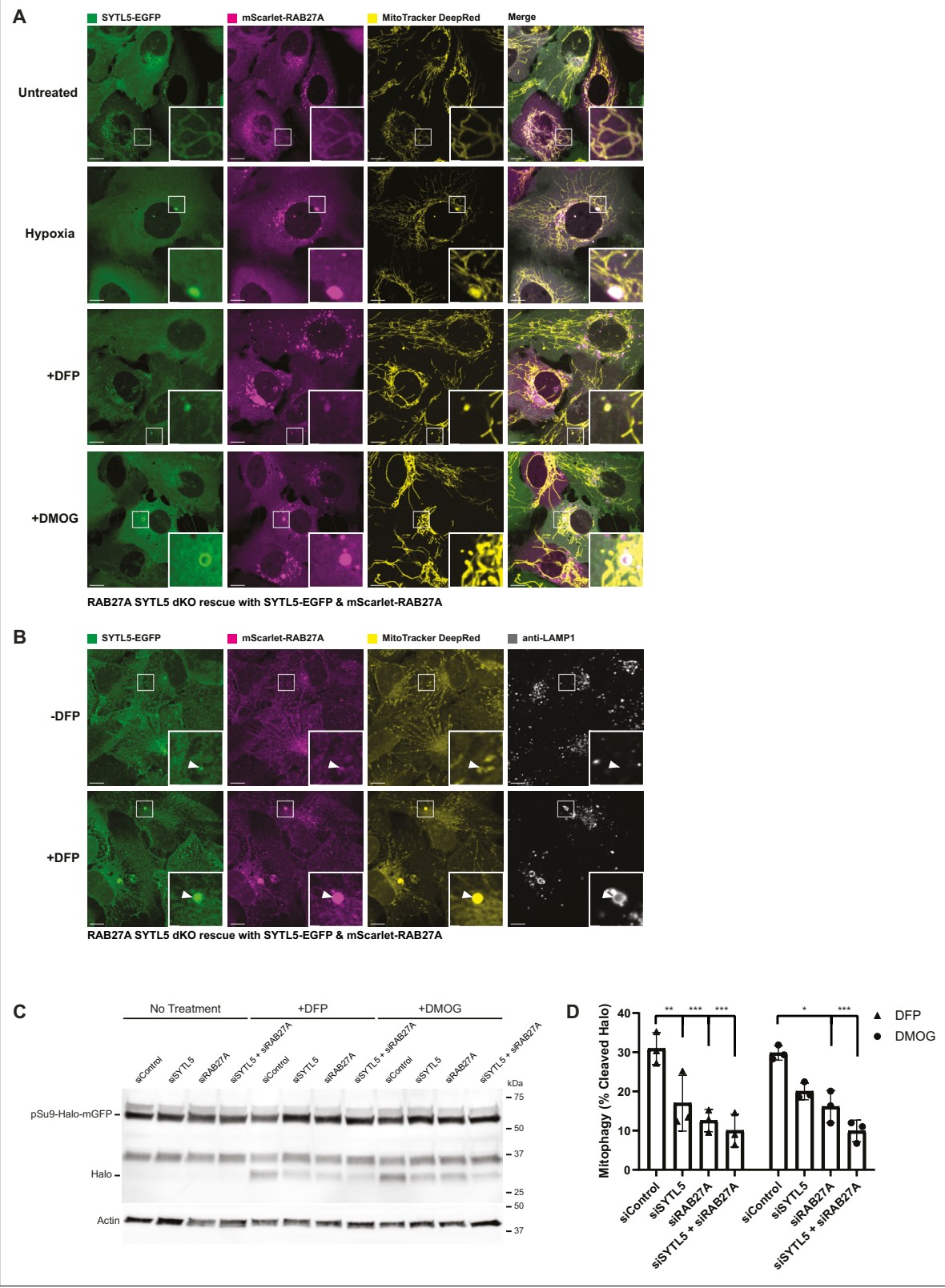

**Figure 4.** Mitochondrial localisation of SYTL5 and RAB27A is not perturbed by autophagy inhibition or mitophagy stimulation. (**A**) SYTL5/RAB27A dKO U2OS cells rescued with SYTL5-EGFP and mScarlet-RAB27A were untreated (upper panel) or treated with 1 mM DFP (2nd panel), hypoxia (1% O₂) (3rd panel), or 1 mM DMOG (4th panel) for 24 hr. All cells were stained with 50 nM MitoTracker DR 30 min prior to live confocal microscopy imaging. Scale bars: 10 µm, 2 µm (insets). (**B**) SYTL5/RAB27A dKO U2OS cells rescued with SYTL5-EGFP and mScarlet-RAB27A were treated or not with 1 mM

*Figure 4 continued on next page*

*Figure 4 continued*

DFP for 24 hr. Cells were stained with 50 nM MitoTracker DR prior to fixation. Fixed cells were stained with a LAMP1 antibody and imaged by confocal microscopy. Scale bars: 10 µm, 2 µm (insets). Arrowheads point to SYTL5/RAB27A-positive structures. (**C**) U2OS cells expressing pSu9-Halo-mGFP were transfected with 20 nM control siRNA, SYTL5 siRNA #1 or RAB27A siRNA #1 for 48 hr before 20 min incubation with the TMR-conjugated Halo ligand followed by 24 hr incubation with 1 mM DFP or 1 mM DMOG. Cell lysates were analysed by western blot for the Halo tag. Actin was used as a loading control. (**D**) Quantification of data in C. Error bars represent the mean with standard deviation between replicates (n=3). Significance was determined by two-way ANOVA followed by Tukey's multiple comparison test. *=p < 0.05, **=p < 0.01, ***=p < 0.001.

The online version of this article includes the following source data and figure supplement(s) for figure 4:

**Source data 1.** Original image files for *Figure 4A*.

**Source data 2.** Original image files for *Figure 4B*.

**Source data 3.** Original uncropped blots for *Figure 2C*.

**Source data 4.** Uncropped blots with the relevant bands clearly labelled for *Figure 2C* and quantification of blots in 2D.

**Figure supplement 1.** SYTL5/RAB27A-positive mitochondrial structures co-localise with autolysosomal markers.

**Figure supplement 1—source data 1.** Original image files for *Figure 4—figure supplement 1A*.

**Figure supplement 1—source data 2.** Original image files for *Figure 4—figure supplement 1B*.

**Figure supplement 1—source data 3.** Original image files for *Figure 4—figure supplement 1B, C*.

**Figure supplement 2.** SYTL5 and RAB27A are dispensable for macro-mitophagy.

**Figure supplement 2—source data 1.** Original uncropped blots for *Figure 4—figure supplement 2F*.

**Figure supplement 2—source data 2.** Uncropped blots with the relevant bands clearly labelled for *Figure 4—figure supplement 2F*.

**Figure supplement 2—source data 3.** Values plotted in *Figure 4—figure supplement 2A, B, C, D, E, F*.

stable expression of Parkin treated with the uncoupler carbonyl cyanide m-chlorophenyl hydrazone (CCCP) to induce Parkin-dependent mitophagy (*Figure 4—figure supplement 2B–E*), together indicating that macro-mitophagy proceeds independently of SYTL5 and RAB27A. In line with this, the starvation-induced autophagic flux was not affected in cells lacking SYTL5 (*Figure 4—figure supplement 2F*).

Taken together, our results indicate that SYTL5 and RAB27A function to regulate the lysosomal turnover of selected mitochondrial proteins in response to hypoxic conditions.

## SYTL5 regulates mitochondrial respiration

Our finding of mitochondrial material within SYTL5-RAB27A-positive structures upon induction of mitochondrial stress through DFP and hypoxia (*Figure 4A*), together with reduced turnover of the Fo-ATPase subunit 9 upon knockdown of SYTL5 and RAB27A (*Figure 4C–D*) and the identification of several proteins involved in the cellular response to stress (*Figure 3C*), including mitochondrial proteins (e.g. DECR1 and TIMM13), in the SYTL5 interactome (*Figure 3A*), led us to investigate a potential role of SYTL5 and RAB27A in mitochondrial function and bioenergetics.

The cellular oxygen consumption rate (OCR) can be measured with the Seahorse analyser upon sequential addition of different inhibitors (oligomycin, CCCP, antimycin and rotenone) that block the function of specific electron transport chain (ETC) complexes (*Hill et al., 2012*). SYTL5 KO cells had a significant decrease in both basal and ATP-linked respiration compared to WT U2OS cells (*Figure 5A–B*), indicating reduced mitochondrial OXPHOS activity. As for RAB27A, both basal respiration and ATP-linked OCR were significantly reduced in cells depleted of RAB27A (RAB27A KO and RAB27A/SYTL5 dKO) relative to the control (*Figure 5A–B*), indicating an important function of these proteins in regulation of mitochondrial respiration.

To test whether the decreased respiration in cells depleted of SYTL5 and/or RAB27A is coupled to a shift to glycolysis, we performed Seahorse analysis of the extracellular acidification rate (ECAR) in cells upon the sequential addition of glucose (increases glycolytic capacity), oligomycin (inhibits mitochondrial ATP production), and 2-DG (a glycolysis inhibitor which causes an abrupt decrease in ECAR). Intriguingly, glycolysis was significantly increased in SYTL5 KO cells as compared to control (*Figure 5C–D*), indicating a possible metabolic shift to compensate for the decrease in OCR observed when SYTL5 was depleted (*Figure 5A–B*). In contrast, ECAR was reduced in cells lacking RAB27A compared to the control, which may indicate an impairment in glucose uptake (*Figure 5C*) and therefore in glycolysis (*Figure 5D*). After adding oligomycin, the ECAR increased in all cell lines, but in

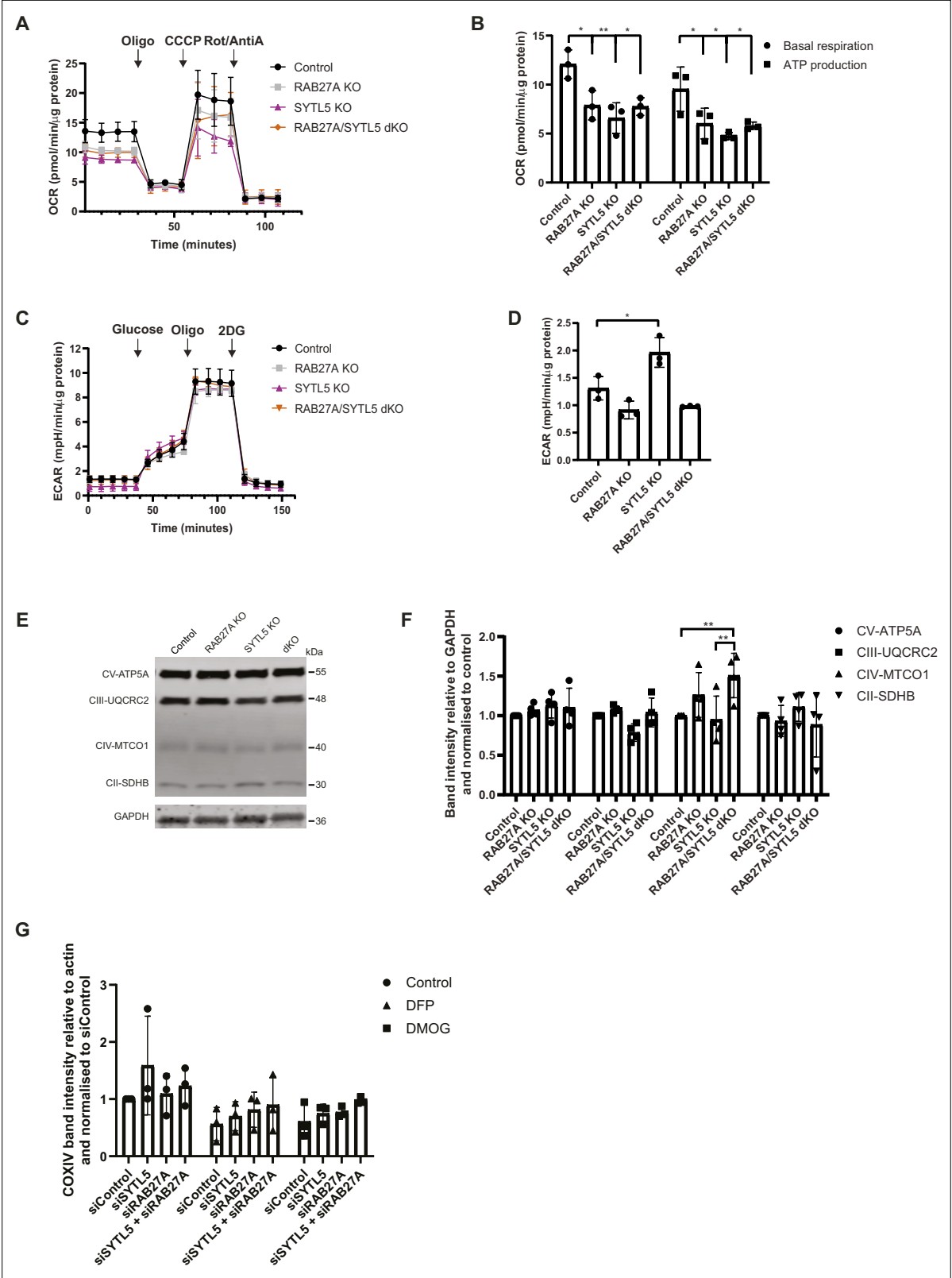

**Figure 5.** SYTL5 and RAB27A regulate mitochondrial bioenergetics. (**A**) Mitochondrial oxygen consumption rate was analysed in U2OS control cells, SYTL5 KO, RAB27A KO, and dKO cells using the Seahorse XF analyser. OCR was measured after sequential addition of oligomycin, CCCP and a combination of rotenone and antimycin-A into the assay medium. Error bars represent the mean with standard deviation between replicates (n=3). (**B**) The four basal OCR measurements per well (before oligomycin addition) were averaged to obtain the basal OCR value and the non-mitochondrial

*Figure 5 continued on next page*

*Figure 5 continued*

respiration was subtracted to determine the basal respiration associated with each condition. The ATP production was calculated by subtracting the proton leak from the maximal respiratory capacity. Error bars represent the mean with standard deviation between replicates (n=3). Significance was determined by two-way ANOVA followed by Bonferroni multiple comparison test, *=$p < 0.05$, **=$p < 0.01$. (**C**) The extracellular acidification rate (ECAR) was analysed in U2OS control cells, SYTL5 KO, RAB27A KO, and dKO cells using the Seahorse XF analyser. ECAR was measured upon addition of glucose, oligomycin, and 2-DG using control and SYTL5-depleted cells. Error bars represent the mean with standard deviation between replicates (n=3). (**D**) The glycolysis rate of each condition was calculated by subtracting the ECAR value after 2-DG treatment from the ECAR after addition of glucose (Glycolysis = ECAR after addition of glucose − ECAR after 2-DG treatment). Error bars represent the mean with standard deviation between replicates (n=3). Significance was determined by one-way ANOVA followed by Tukey multiple comparison test, *=$p < 0.05$. (**E**) Cell lysates from U2OS control, RAB27A KO, SYTL5 KO, and dKO cells were analysed by western blot for the indicated electron transport chain complex-II, -III, -V, and -V proteins. GAPDH was used as a loading control. (**F**) Quantification of data in E. Band intensities were quantified relative to GAPDH and normalised to control. Error bars represent the mean with standard deviation between replicates (n=4). Significance was determined by two-way ANOVA followed by Bonferroni multiple comparison test, **=$p < 0.01$. (**G**) U2OS cells were transfected with 20 nM control siRNA, SYTL5 siRNA #1, or RAB27A siRNA #1 for 48 hr followed by 24 hr incubation with 1 mM DFP or 1 mM DMOG. Cell lysates were analysed by western blot for COXIV. Actin was used as a loading control. Error bars represent the mean with standard deviation between replicates (n=3).

The online version of this article includes the following source data for figure 5:

**Source data 1.** Original uncropped blots for *Figure 5E*.

**Source data 2.** Uncropped blots with the relevant bands clearly labelled for *Figure 5E*.

**Source data 3.** Values plotted in *Figure 5B, D, F and G*.

---

RAB27A KO and dKO cell lines, the increase was reduced when compared to the control cell line (*Figure 5C–D*).

The clear reduction in mitochondrial respiration observed in cells lacking RAB27A and SYTL5 led us to investigate the abundance of various ETC proteins. Depletion of SYTL5 and/or RAB27A did not affect the overall abundance of ATP5A (complex V), UQCRC2 (complex III), or SDHB (complex II), but we observed a significant increase in the level of MTCO1 (complex IV) in cells lacking RAB27A (*Figure 5E–F*). There was, however, no significant difference in the level of COXIV, another subunit of the ETC complex IV, upon siRNA-mediated knockdown of SYTL5, RAB27A or both compared to siControl in basal condition or upon induction of mitophagy with DFP or DMOG (*Figure 5G*).

Taken together, our data show that both RAB27A and SYTL5 are important for normal mitochondrial respiration and ATP production, while SYTL5 seems to regulate a metabolic switch to glycolysis.

## Low SYTL5 expression is related to reduced survival for adrenocortical carcinoma patients

Since our data indicate that SYTL5 may be a regulator of the metabolic switch from oxidative phosphorylation to glycolysis (the Warburg effect; *Figure 5A–D*) that is frequently associated with cancer cells, we decided to analyse publicly available expression datasets to explore a possible link between SYTL5 and cancer. Using bulk RNA expression data from healthy/normal tissue samples in the GTEx project, we found that SYTL5 is preferentially expressed in adrenal gland and various regions of the brain (*Figure 6A*). Adrenocortical carcinoma (ACC) is a rare type of aggressive cancer with poor prognosis and limited treatment options that develops in the adrenal cortex (*Else et al., 2014*). A comparison of gene expression levels (*Tang et al., 2017*) in normal adrenal gland samples and ACC samples showed that SYTL5 mRNA expression is significantly reduced in tumour samples (*Figure 6B*). Furthermore, patient survival analysis using data obtained from the TCGA-ACC cohort indicates that low SYTL5 mRNA expression is associated with a significantly reduced survival of ACC patients (*Figure 6C*), suggesting a possible tumour suppressor function for SYTL5 in ACC cancer progression. RAB27A expression levels were also reduced in tumour tissue compared to normal samples, although to a lesser extent than SYTL5 (*Figure 6D*), and overall patient survival in ACC patients did not show any relationship with RAB27A expression levels (*Figure 6E*).

The adrenal gland cortex secretes several steroid hormones, such as mineralocorticoids (e.g. aldosterone), glucocorticoids (e.g. cortisol), and adrenal androgens (DHEA and DHEA-sulphate; *Wang and Rainey, 2012*), to the bloodstream. ACC tumours are characterised by excess adrenocortical hormone production in nearly 45–70% of patients, with hypercortisolism being the most common (*Else et al., 2014*). As mitochondria contain enzymes essential for steroid hormone biosynthesis from cholesterol (*Miller, 2013*), we asked whether SYTL5 may regulate cortisol production. SYTL5 or RAB27A was

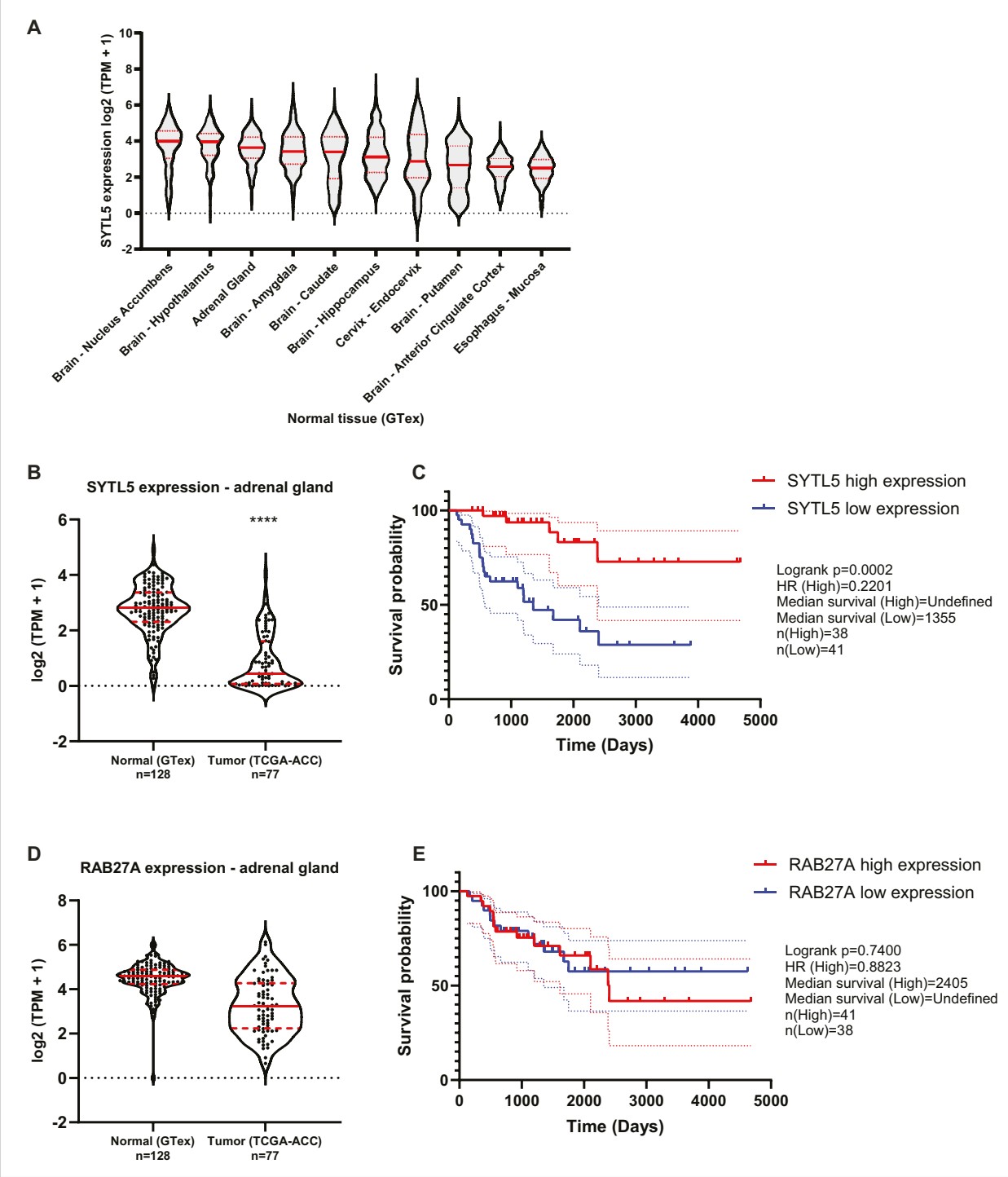

**Figure 6.** Low SYTL5 expression is related to reduced survival for adrenocortical carcinoma patients. (**A**) SYTL5 expression levels in normal/healthy tissues, using data from the GTEx project (**v8**). Shown are the 10 tissue sites for which SYTL5 is most highly expressed (i.e. based on median expression levels). TPM = transcripts per million. Median is represented by a red solid line and quartiles by red dashed lines. (**B**) Comparison of SYTL5 gene expression levels in normal adrenal gland samples (GTEx, n=128) and adrenocortical carcinoma samples (The Cancer Genome Atlas (TCGA) ACC cohort, n=77). Median expression is represented by a red solid line and quartiles by red dashed lines. Statistical analysis was performed using the unpaired t-test with Welch's correction. TPM = transcripts per million. (**C**) Kaplan-Meier plot for ACC patient survival related to SYTL5 expression levels. The events related to high SYTL5 expression (expression level above median) are indicated in red, and the dotted lines represent 95% confidence interval. Events related to low SYTL5 expression (expression level below median) are indicated in blue, and the dotted lines represent 95% confidence interval. Statistical analysis was performed using the Logrank (Mantel-Cox) test. (**D**) Comparison of RAB27A gene expression levels in normal adrenal

*Figure 6 continued on next page*

*Figure 6 continued*

gland samples (GTEx, n=128) and adrenocortical carcinoma samples (The Cancer Genome Atlas (TCGA) ACC cohort, n=77). Median expression is represented by a red solid line and quartiles by red dashed lines. Statistical analysis was performed using the unpaired t-test with Welch's correction. (**E**) Kaplan-Meier plot for ACC patient survival related to RAB27A expression levels. The events related to high RAB27A expression (expression level above median) are indicated in red, and the dotted lines represent 95% confidence interval. Events related to low RAB27A expression (expression level below median) are indicated in blue, and the dotted lines represent 95% confidence interval. Statistical analysis was performed using the Logrank (Mantel-Cox) test.

The online version of this article includes the following source data and figure supplement(s) for figure 6:

**Source data 1.** Values plotted in *Figure 6A, B, C, D and E*.

**Figure supplement 1.** NCl-H295R cells were transfected with 20 nM control siRNA, SYTL5 siRNA or RAB27A siRNA oligos for 72 h.

**Figure supplement 1—source data 1.** Values plotted in *Figure 6—figure supplement 1*.

depleted by siRNA in the ACC cell line H295R, followed by quantification of cortisol secretion using a colourimetric assay. We did not observe a significant change in cortisol secretion upon SYTL5 or RAB27A knockdown (*Figure 6—figure supplement 1*), indicating that the correlation between low SYTL5 expression levels and ACC patient survival likely is unrelated to effects on cortisol production.

## Discussion

SYTL5 has been identified as a RAB27A effector protein with affinity for membranes through its C2 lipid-binding domains (*Kuroda et al., 2002a*), but nothing is known about a potential role of SYTL5 in regulation of RAB27A-dependent membrane trafficking pathways. Using the osteosarcoma-derived U2OS cell line, here we confirm that SYTL5 interacts with RAB27A and show that it localises to mitochondria and small, highly mobile vesicles interacting with the mitochondrial network, as well as to endolysosomal and endocytic compartments, indicating a role of SYTL5 in cellular membrane trafficking events.

Most interestingly, mitochondrial recruitment of SYTL5 involves coincidence detection, as it requires both its RAB27-binding SHD domain and its lipid-binding C2 domains. Indeed, we demonstrate that SYTL5 is recruited to mitochondria in a RAB27A-dependent manner and that RAB27A itself is localised to mitochondria in a manner dependent on its GTPase activity, as both RAB27A GDP and GTP mutants (T23N and Q78L, respectively) showed a dispersed cytoplasmic localisation. This is in line with a study showing that the GTPase activity of RAB27A is required for its localisation to melanosomes (*Ishida et al., 2014*). The mechanisms involved in mitochondrial recruitment of RAB27A are not clear. It is, however, interesting to note that other RAB proteins are recruited to the mitochondria by regulation of their GTPase activity. During parkin-mediated mitophagy, RABGEF1 (a guanine nucleotide exchange factor) is recruited through its ubiquitin-binding domain and directs mitochondrial localisation of RAB5, which subsequently leads to recruitment of RAB7 by the MON1/CCZ1 GEF complex (*Yamano et al., 2018*). Intriguingly, ubiquitination of the RAB27A GTPase activating protein alpha (TBC1D10A) is reduced in the brain of Parkin KO mice compared to controls (*Key et al., 2019*), suggesting a possible connection of RAB27A with regulatory mechanisms that are linked with mitochondrial damage and dysfunction.

SYTL5 and RAB27A are both members of protein families, suggesting possible functional redundancies from RAB27B or one of the other SYTL isoforms. While RAB27B has a very low expression in U2OS cells, all five SYTLs are expressed. However, when knocking out or knocking down SYTL5 and RAB27A, we observe significant effects that we presume would be negated if their isoforms were providing functional redundancies. Moreover, we did not detect any other SYTL protein or RAB27B in the SYTL5 interactome, confirming that they do not form a complex with SYTL5.

Given the presence of mitochondria-associated small and highly mobile SYTL5/RAB27A-positive vesicles that contain mitochondrial material and stain positive for LAMP1, we speculate that RAB27A and SYTL5 facilitate mitophagy-mediated lysosomal delivery of mitochondria or selected mitochondrial components. Indeed, SYTL5 and RAB27A-positive vesicles containing mitochondrial material were observed in cells upon hypoxia or hypoxia-mimicking (DFP and DMOG) treatments, where some vesicles stained positive for the autophagy markers p62 and LC3B upon co-treatment with the lysosomal V-ATPase inhibitor BafA1. Intriguingly, in cells treated with DFP and DMOG, we observed a significant decrease in lysosomal cleavage of the mitophagy reporter pSu9-Halo-mGFP (*Yim et al.,*

*2022*) upon depletion of SYTL5 and/or RAB27A. Additionally, when blotting for various endogenous proteins of the electron transport chain complexes, we found that the level of the cytochrome c oxidase (complex IV) subunit MTCO1/COX1 was significantly increased in cells lacking RAB27A. However, neither the basal level of COXIV, another complex IV subunit, nor the lysosomal targeting of a mitochondrial matrix reporter (IMLS) was affected by depletion of SYTL5 and/or RAB27A. Expression of the mitophagy receptor BNIP3L, a HIF-1a target, was not affected by SYTL5 and/or RAB27A depletion. Mitochondrial recruitment of SYTL5 and RAB27A was also independent of the core autophagy machinery components ULK1 and VPS34. Taken together, our data indicate that mitochondrial RAB27A/SYTL5 functions as positive regulators of selective clearance of mitochondrial components in a piece-meal mitophagy-dependent manner.

The role of SYTL5 and RAB27A in the turnover of proteins involved in mitochondrial oxidative phosphorylation and ATP production is reminiscent of the recently described VDIM (vesicles derived from the inner mitochondrial membrane) pathway (*Prashar et al., 2024*). VDIMs are formed by IMM herniation through pores in the outer mitochondrial membrane, followed by their engulfment by lysosomes in proximity to mitochondria in a process that is independent of the core autophagy machinery. However, in contrast to our findings, VDIMs lack LC3 and p62, and oxidative stress-induced VDIM formation leads to selective degradation of IMM proteins, including COX4.

Mitochondria are known stress sensors and hubs for cellular adaptation (*Montava-Garriga and Ganley, 2020*). When analysing the interactome of SYTL5, we found that many candidates were related to the cellular response to stress and oxygen-containing compounds, as well as vesicle-mediated transport and secretion. In line with a role for SYTL5/RAB27A in the regulation of mitochondrial stress, we found that cells lacking SYTL5 and/or RAB27A displayed reduced OXPHOS activity and ATP production. A shift from OXPHOS to anaerobic glycolysis occurs in normal cells during hypoxic conditions and often in cancer cells even under normoxic conditions, commonly referred to as the Warburg effect (*Liberti and Locasale, 2016*). Intriguingly, while the depletion of SYTL5 triggered a shift to glycolysis as observed by an increase in the extracellular acidification rate (ECAR) due to cellular lactate production, depletion of RAB27A or both RAB27A/SYTL5 had no significant effect, indicating that SYTL5 may function as a negative regulator of RAB27A and subsequently the Warburg effect.

The observed switch from OXPHOS to glycolysis in U2OS cells depleted of SYTL5 prompted us to look at cancer patient databases. We observed, using data from GTEx, that the adrenal gland is among the tissues with highest levels of SYTL5 gene expression, and that its expression is reduced in adrenocortical carcinoma samples (TCGA-ACC) when compared to normal adrenal gland tissue. Moreover, low SYTL5 expression is associated with reduced ACC patient survival. One characteristic of this type of cancer is an increase in steroid hormone production and secretion. Most steroid hormones are produced in mitochondria (*Miller, 2013*), but we did not observe any changes in the level of cortisol in ACC cells depleted of SYTL5. The mechanisms linking SYTL5 to the Warburg effect and ACC need further investigation, but it is interesting to note that RAB27A expression levels do not correlate with ACC survival.

To summarise, we show that SYTL5 is a RAB27A effector protein being recruited to mitochondria in a RAB27A-dependent manner. Upon hypoxia, RAB27A and SYTL5 promote lysosomal clearance of selected mitochondrial components, and both proteins are needed for mitochondrial respiration. We identify SYTL5 as a negative regulator of the Warburg effect and potentially tumourigenesis.

# Materials and methods

## Key resources table

| Reagent type (species) or resource | Designation | Source or reference | Identifiers | Additional information |
|---|---|---|---|---|
| Cell line (*Homo sapiens*) | U2OS | ATCC | HTB-96 | |
| Cell line (*Homo sapiens*) | HEK-FT | Invitrogen | R70007 | |
| Cell line (*Homo sapiens*) | NCI-H295R | ATCC | CRL-2128 | |
| Sequence-based reagent | siRNA to SYTL5 | Thermo Fisher Scientific | s41276 | Silencer Select |
| Sequence-based reagent | siRNA to SYTL5 | Thermo Fisher Scientific | s41277 | Silencer Select |

*Continued on next page*

*Continued*

| Reagent type (species) or resource | Designation | Source or reference | Identifiers | Additional information |
|---|---|---|---|---|
| Sequence-based reagent | siRNA to SYTL5 | Thermo Fisher Scientific | s41275 | Silencer Select |
| Sequence-based reagent | siRNA to RAB27A | Thermo Fisher Scientific | s11695 | Silencer Select |
| Transfected construct (human) | siRNA to RAB27A | Thermo Fisher Scientific | s532296 | Silencer Select |
| Antibody | Mouse monoclonal α-Tubulin | Sigma-Aldrich | #T5168 | 1:20,000 |
| Antibody | Mouse monoclonal β-Actin | Cell signalling technology | #3700 | 1:5000 |
| Antibody | Rabbit monoclonal BNIP3L | Cell signalling technology | #12396 | 1:1000 |
| Antibody | Rabbit monoclonal COXIV | Cell signalling technology | #4850 | 1:1000 |
| Antibody | Mouse monoclonal EGFP | Clontech | #632569 | 1:1000 |
| Antibody | Rabbit polyclonal GFP | Abcam; | #ab290 | 1:1000 |
| Antibody | Rabbit monoclonal FLAG Tag | Cell signalling technology | #14793 | 1:500 |
| Antibody | Rabbit monoclonal GAPDH | Cell signalling technology | #5174 | 1:1000 |
| Antibody | Mouse monoclonal Halo Tag | Promega | #G9211 | 1:1000 |
| Antibody | Mouse monoclonal LAMP1 | Santa Cruz Biotechnology | #sc-20011 | 1:2000 |
| Antibody | Rabbit polyclonal LC3 | MBL | PM036 | 1:500 |
| Antibody | Mouse monoclonal OXPHOS | Abcam | #ab110413 | Cocktail of 5mAbs 1:500 |
| Antibody | Guinea pig polyclonal p62 | Progen | #GP62-C | 1:2000 |
| Antibody | Mouse monoclonal RAB27A | Santa Cruz Biotechnology | #sc-74586 | 1:1000 |
| Antibody | Rabbit polyclonal SYTL5 | Sigma-Aldrich | #HPA026074 | 1:1000 |
| Antibody | Mouse monoclonal TIM23 | BD Biosciences | #611223 | 1:1000 |
| Antibody | Mouse monoclonal TOMM20 | Santa Cruz Biotechnology | #sc-17764 | 1:1000 |
| Recombinant DNA reagent | pcDNA5/FRT/TO_SYTL5-EGFP | This paper | | See methods |
| Recombinant DNA reagent | pLVX-SV40-mScarlet-RAB27A | This paper | | See methods |
| Recombinant DNA reagent | pLVX-SV40-mScarlet-RAB27A (T23N) | This paper | | See methods |
| Recombinant DNA reagent | pLVX-SV40-mScarlet-RAB27A (Q78L) | This paper | | See methods |
| Recombinant DNA reagent | pLVX-CMV-SYTL5-EGFP-3xHA | This paper | | See methods |
| Recombinant DNA reagent | pLVX-CMV-SYTL5-EGFP-3xFLAG | This paper | | See methods |
| Recombinant DNA reagent | pLVX-CMV-SYTL5 (ΔC2AB)-EGFP-3xFLAG | This paper | | See methods |
| Recombinant DNA reagent | pLVX-CMV-SYTL5 (ΔSHD)-EGFP-3xFLAG | This paper | | See methods |
| Commercial assay or kit | FLAG immuno-precipitation kit | Sigma-Aldrich | #FLAGIPT-1 | |
| Commercial assay or kit | Cortisol Immunoassay kit | R&D Systems | #KGE008B | |
| Chemical compound, drug | DFP | Sigma-Aldrich | #379409 | |
| Chemical compound, drug | DMOG | Sigma-Aldrich | #D3695 | |
| Chemical compound, drug | VPS34IN1 | Selleckchem | # S7980 | |
| Chemical compound, drug | MRT68291 | Selleckchem | #S7949 | |

*Continued on next page*

*Continued*

| Reagent type (species) or resource | Designation | Source or reference | Identifiers | Additional information |
|---|---|---|---|---|
| Chemical compound, drug | Oligomycin A | SelleckChem | #S1478 | |
| Chemical compound, drug | CCCP | Enzo Life Sciences | #BML-CM124-0500 | |
| Chemical compound, drug | Rotenone | Sigma-Aldrich | #R8875 | |
| Chemical compound, drug | Antimycin | Sigma-Aldrich | #A8674 | |
| Software, algorithm | Prism | GraphPad | | Statistical analysis |
| Other | Hoechst | Thermo Fisher Scientific | #H1399 | |
| Other | LysoTracker Red DND-99 | Thermo Fisher Scientific | #L7528 | |
| Other | MitoTracker Deep Red FM | Thermo Fisher Scientific | # M22426 | |
| Other | MitoTracker Green FM | Thermo Fisher Scientific | #M7514 | |
| Other | MitoTracker Red CMXRos | Thermo Fisher Scientific | #M7512 | |

## Cell culture, lentivirus production, and stable cell line generation

Human bone osteosarcoma epithelial cells (U2OS FlpIn TRex cell line, kindly provided by Steve Blacklow, Harvard Medical School, US) were grown in complete medium of Dulbecco's Modified Eagle Medium (DMEM; Lonza #12–604 F and Gibco # 31966047) with 10% v/v foetal bovine serum (FBS; Sigma-Aldrich #F7524), 100 U/ml Penicillin and 100 µg/ml Streptomycin (P/S; Thermo Fisher Scientific #15140122) at 37 °C with 5% $CO_2$.

For lentiviral particle production, HeK-FT cells were co-transfected with 1.6 µg of pLVX lentiviral vector, pCMV-VSV-G and psPAX2 using X-tremeGENE 9 DNA Transfection Reagent (Sigma #XTG9-RO). For virus infection, cells were seeded in complete medium and supplemented with 8 µg/ml poly-brene (Santa Cruz Biotechnology #sc-134220). 2 µg/ml Puromycin (Sigma Aldrich #P7255) was added after 24 hr post-infection for selection.

To generate U2OS stably expressing SYTL5-EGFP, U2OS FlpIn TRex cells and U2OS FlpIn TRex_mScarlet-RAB27A were co-transfected with 0.1 µg pcDNA5/FRT/TO_SYTL5-EGFP and 0.9 µg pOG44 Flp recombinase expression vector using Lipofectamine 2000 (Thermo Fisher Scientific #11668019) according to the manufacturer's instructions. To produce stable cell lines expressing pLVX-SV40-mScarlet-RAB27A(T23N/Q78L) and pLVX-CMV-SYTL5-EGFP-3xHA, U2OS dKO cells were infected with lentiviral particles followed by fluorescence-activated cell sorting (FACS) for double mScarlet/EGFP expressing cells. To generate stable cells expressing pLVX-CMV-SYTL5-EGFP-3xFLAG, pLVX-CMV-SYTL5 (ΔC2AB)-EGFP-3xFLAG and pLVX-CMV-SYTL5 (ΔSHD)-EGFP-3xFLAG, U2OS FlpIn TRex cells were infected with lentiviral particles. To generate U2OS stably expressing pSu9-Halo-mGFP, U2OS cells were infected with retroviral particles generated using pUMVC (Addgene # 8449) and pMRX-IB-pSu9-HaloTag7-mGFP (Addgene # 184905). Stable cell lines were maintained in complete medium with additional 5 µg/ml blasticidin S (Thermo Fisher Scientific #R21001) or 2 µg/ml puromycin (Sigma-Aldrich #P7255). For starvation experiments, cells were washed with PBS and incubated for 4 hr in Earle's balanced salt solution (EBSS; Gibco #24010043).

NCI-H295R cells were obtained from ATCC (#CRL-2128) and maintained in DMEM: F12, HEPES medium (Gibco #11330032) supplemented with 2.5% Nu-Serum (Corning #355100), 1% ITS +Premix Universal culture supplement (Corning #354352), and 100 µg/ml P/S (Thermo Fisher Scientific #15140122) at 37 °C with 5% $CO_2$.

All cell lines were tested regularly for mycoplasma, and the result was negative. Authentication was by STR profiling.

## Hypoxia and drug treatments

After cells were seeded, they were transferred to an INVIVO$_2$ 200 Hypoxic workstation (Ruskinn). Hypoxia treatment was carried out for 24 hr with oxygen concentration set to 1%.

Drugs were added into the cell media. Treatment durations and drug concentrations are indicated in the figure legends. Drugs used were DFP (Sigma-Aldrich #379409), DMOG (Sigma-Aldrich #D3695), VPS34IN1 (Selleckchem # S7980), and MRT68291 (Selleckchem #S7949).

## Molecular cloning

SYTL5 was amplified from WT U2OS cDNA (5´-ATGTCTAAGAACTCAGAGTTCATC-3´ and 5´-TTAG AGCCTACATTTTCCCATG-3´) and cloned into Zero Blunt TOPO vector using Zero Blunt TOPO PCR Cloning Kit (Thermo Fisher Scientific #450245) according to the manufacturer´s instructions. pcDNA5/ FRT/TO_SYTL5-EGFP was generated by Gibson Assembly using the TOPO-SYTL5 vector as a template for SYTL5 and cloned into a pcDNA5/FRT Mammalian Expression Vector (Thermo Fisher Scientific #V601020) or into a pLVX-CMV lentiviral expression vector. Truncated SYTL5 (ΔSHD) and SYTL5 (ΔC2AB) were generated by Gibson assembly using the pcDNA5/FRT/TO_SYTL5-EGFP construct as a template and cloned into a pLVX-CMV lentiviral expression vector.

RAB27A was amplified from WT U2OS cDNA and cloned into a pLVX-SV40 lentiviral expression vector. RAB27A mutations were performed using the site-directed mutagenesis QuikChange II kit (Agilent #200524) according to the manufacturer's instructions using the following primer pairs:

| Mutation | Forward primer | Reverse primer |
| --- | --- | --- |
| RAB27A_T23N (c68a) | atattggtaaagtacactgttcttccctacaccagagtc | gactctggtgtagggaagaacagtgtactttaccaatat |
| RAB27A_Q78L (a233t) | tacgaaacctctccagccctgctgtgtcc | ggacacagcagggctggagaggtttcgta |

## CRISPR knockout cell line generation and validation

SYTL5 and RAB27A gene knockouts were performed as previously described by *Ran et al., 2013*; . Briefly, gRNA target sequences were designed using Wellcome Sanger Institute Genome Editing (WGE) design tool available at https://wge.stemcell.sanger.ac.uk/find_crisprs.

| Target gene | gRNA sequence | Genomic location |
| --- | --- | --- |
| RAB27A | AGTGGCTCCATCCGGCCCAC | chr15:55230454 |
| SYTL5 guide 1 | CGACTAATACTTGCCAGAGC | chrX:38034292 |
| SYTL5 guide 2 | ATACGTAGAACAGCCTCCAC | chrX:38033693 |

DNA oligos were annealed and phosphorylated before cloning into pSpCas9(BB)–2A-Puro (PX459) V2.0 (addgene # 62988). The obtained CRISPR plasmids were transfected into U2OS Flp-In TRex cells using Lipofectamine 2000 according to the manufacturer's instructions and selection with 2 µg/ml puromycin (Sigma-Aldrich #P7255) was applied for 24 hr. Limiting dilutions were performed using a concentration of 15 cells/ml and seeded in complete medium in a 384-well plate. Single colonies were collected ~10 days after seeding and expanded for a further 3 weeks. For genotyping, gDNA from each single clone line was isolated using QIAamp DNA Mini Kit (QIAGEN #51304) and the targeted genomic region was amplified by PCR using flanking primers (RAB27A: 5´-ATAACCTCTCCCTTGACCTTGTATG-3´ and 5´-TAGATGCCTTTGGGATTTGTACTGA-3´; SYTL5: 5´-CAGGTCCCTTTCTTCTCGCA-3´ and 5´-TCAGTCAGCTGCAAGAGTGG-3´). The PCR product was cloned into the Zero Blunt TOPO vector using Zero Blunt TOPO PCR Cloning Kit according to the manufacturer's instructions. RAB27A clonal lines were validated by western blot, and SYTL5 deletions were detected by running PCR products in 1% w/v agarose gel (Thermo Fisher Scientific #R2801).

## siRNA knockdown by reverse transfection

For siRNA knockdown, cells were incubated with OptiMEM (Thermo Fisher Scientific # 31985047) and Lipofectamine RNAiMAX (Thermo Fisher Scientific #13778150) for 5 min at room temperature (RT). Followed by the addition of 20 nM siRNA diluted in OptiMEM and a 15 min incubation at RT. Cells were fixed/harvested after 72 hr of knockdown. The siRNA used for mRNA depletion is the following: siControl: 5´-UAACGACGCGACGACGUAAtt-3´; siSYTL5 #1: 5´-CUCUUAGAAGCAAAACGUAtt-3´; siSYTL5 #2: 5´-CAACAAGCGUAAGACCAAAtt-3´; siSYTL5 #3: 5'-GGUUUGUGCUUCAACCCAAtt-3'; siRAB27A #1: 5´-GGAAGACCAGUGUACUUUAtt-3´ and siRAB27A #2: 5´-CCAGUGUACUUUACCA AUAtt-3´.

## RNA isolation, cDNA synthesis, and RT-PCR

RNA was isolated using TRIzol reagent (Thermo Fisher Scientific #15596026) and cDNA was synthesised using SuperScript III Reverse Transcriptase (Thermo Fisher Scientific #18080085) according to the manufacturer's instructions. qPCR analysis was performed using KAPA SYBR FAST qPCR Kit (Sigma-Aldrich #KK4601) and relative mRNA expression levels were normalised to the expression levels of TATA-binding protein (TBP) using the comparative ΔCt method. The primers used were the following:

SYTL5: 5′-GTGACAAAATCGCGCAGCTA-3′, 5′-GGACAACATCAGTGCCGAGA-3′
RAB27A: 5'-GGAGAGGTTTCGTAGCTTAACG-3', 5'-CCACACAGCACTATATCTGGGT-3'
TBP: 5′-CAGAAAGTTCATCCTCTGGGCT-3′, 5′-TATATTCGGCGTTTCGGGCA-3′

## Immunofluorescent staining

Cells were fixed in 4% paraformaldehyde (PFA) (Sigma-Aldrich #158127) in PBS for at least 15 min at 37 °C, quenched for 10 min using 0.05 M NH4Cl in PBS, permeabilised in 0.05% saponin in PBS for 5 min and blocked in 1% BSA in PBS for 30 min at room temperature. Antibody staining was performed in a wet chamber for 1 hr at room temperature, and the antibodies were diluted in PBS containing 0.05% saponin. Nuclei staining was performed using Hoechst 33342 (Thermo Fisher Scientific #H1399) diluted in PBS at 1 µg/ml.

## Light microscopy

Cells expressing inducible fluorescent proteins were induced with 100 ng/ml doxycycline (Clontech #631311) for 24 hr. For live cell confocal microscopy, cells were incubated with indicated dyes prior to imaging. Imaging was carried out with either the Andor Dragonfly High Speed Confocal Microscope or the Nikon Crest V3 with a 60 x oil immersion objective (NA 1.4). High-content widefield imaging was conducted with the ImageXpress Micro Confocal (Molecular Devices) microscope with a 20 x objective (NA 0.45). Fixed immunofluorescence confocal imaging was performed using the Andor Dragonfly High Speed Confocal Microscope or Nikon Crest V3, both with a 60 x oil immersion objective (NA 1.4) or Zeiss LSM 800 microscope (Zen Black 524 2012 SP5 FP3, Zeiss) with a 63 x oil immersion objective (NA 1.4).

## Correlative light electron microscopy

U2OS cells co-expressing SYTL5-EGFP and mScarlet-RAB27A were seeded in gridded glass-bottom cell culture dishes (MatTek #P35G-1.5–14-CGRD). Cells growing in monolayer were fixed in warm (≈37 °C) 3.7% PFA in 0.2 M HEPES (pH 7) and imaged using the Andor Dragonfly High Speed Confocal Microscope. After imaging, the samples were fixed using 2% v/v glutaraldehyde in 0.2 M HEPES, pH 7.4. After post-fixation in 2% v/v osmium tetroxide and 3% v/v K4[Fe(CN)6], samples were embedded in Epon and 60 nm thickness sections were cut with a diamond knife. Sections were analysed using a JEM-1400 Plus Transmission Electron Microscope.

## Western blot

Cells were harvested in RIPA buffer 50 mM Tris pH 7.4, 150 mM NaCl (Merck #1064041000), 0,5% sodium deoxycholate (Sigma-Aldrich #30970), 1% NP-40 (Sigma-Aldrich #I3091), 1 mM EDTA (VWR #20302–293), 0,1% SDS (Sigma-Aldrich #L3771) containing 1 x protease inhibitor (Merck #11697498001) and 1 x phosphatase inhibitor (Merck #4906845001). 20 µg of lysate was loaded on SDS-PAGE gel (BioRad #5671095) and proteins transferred to a PVDF membrane (Merck #IPFL00010) which was blocked using 5 x v/v casein blocking buffer in PBS for 1 hr at room temperature. The obtained membranes were immunoblotted using primary and secondary antibodies with washing steps using PBST (1 x Phosphate-Buffered Saline with 1% Tween20 (Sigma-Aldrich #P1379)). Membranes were analysed using the Odyssey CLx imaging system (Li-cor biosciences), and protein levels were quantified by densitometry with ImageStudio Lite software (Li-cor biosciences).

## Macro-mitophagy analysis

siRNA knockdown was conducted as previously described in U2OS-IMLS cells seeded in complete media supplemented with 100 ng/ml doxycycline. Culture media were replaced 24 hr after transfection, and 1 mM DFP (Sigma-Aldrich #379409) treatment was added after 48 hr. For U2OS-IMLS cells expressing PARKIN, 20 µM CCCP (Enzo Life Sciences #BML-CM124-0500) was used for 16 hr and

10 µM Q-VD-OPh (Sigma-Aldrich #SML0063-1MG) pan-caspase inhibitor was included to support cell survival (*Lazarou et al., 2015*; *Caserta et al., 2003*). Control wells with 100 nM BafA1 (AH diagnostics #BML-CM110-0100) were dosed 2 hr prior to fixation. After 72 hr, cells were washed with PBS pH 7 and fixed in warm (≈37 °C) 3.7% PFA in 0.2 M HEPES (pH 7) for 15 min at 37 °C. After washing with PBS, cells were incubated with 2 µg/ml Hoechst and widefield images were obtained using a high-content imaging microscope. The area of red-only puncta per cell (which represent mitochondria delivered to lysosomes as the EGFP signal is quenched in the acidic lysosome) was quantified using a Cell Profiler pipeline. The addition of BafA1 was used as a control to confirm that the mCherry-only puncta corresponded to lysosomal structures.

## Halo assay for mitophagy flux

siRNA knockdown was conducted as previously described in U2OS cells expressing the pSu9-Halo-mGFP reporter. 48 hr after transfection, culture media was changed to contain 100 nM tetramethyl-rhodamine (TMR)-conjugated Halo ligand (Promega #G8251). After a 20 min incubation, cells were washed twice with PBS and the media replaced with either media containing no treatment, 1 mM DFP or 1 mM DMOG. 72 hr after transfection, cells were harvested and western blot conducted (as described earlier). Membranes were incubated with the Halo Tag antibody and actin antibody. Membranes were imaged using the Chemidoc MP system (Bio-Rad) and band intensity quantified in Image Lab 6.1 (Bio-Rad). Bands were normalised to actin loading control. The percentage cleaved Halo was calculated by the formula (Free Halo / (Free Halo +Full Length)) x 100.

## Oxygen consumption rate measurement by Seahorse XF analyser

U2OS cells were seeded in complete medium at the density of $4x10^4$ cells/well into XF24 cell culture microplates (Agilent #100777–004). The sensor cartridge was hydrated for 24 hr using XF calibrant in a 37 °C non-$CO_2$ incubator. Before analysis, the cell medium was replaced by warm (≈37 °C) DMEM without Sodium Bicarbonate (pH 7.4) and placed for 1 hr in a 37 °C non-$CO_2$ incubator. Each measurement cycle consisted of a mixing time of 3 min and a data acquisition period of 3 min (three data points). OCR results refer to the average rates during each measurement cycle. The final concentration for each mitochondrial inhibitor used in this assay was 1.5 µM Oligomycin A (SelleckChem #S1478), 1 µM CCCP, 0.5 µM Rotenone (Sigma-Aldrich #R8875-1G), and 0.5 µM Antimycin (Sigma-Aldrich #A8674). Four baseline measurements were performed before adding each compound, and three response measurements were taken after each compound was added. Data analysis was performed using Seahorse Analytics software available at https://seahorseanalytics.agilent.com/ and OCR data was normalised to protein concentration in each well.

## Extracellular acidification rate (ECAR) measurement by Seahorse XF analyser

U2OS cells were seeded in complete medium at the density of $4x10^4$ cells/well into XF24 cell culture microplates. The sensor cartridge was hydrated for 24 hr using XF calibrant in a 37 °C non-$CO_2$ incubator. Before analysis, cell medium was replaced by warm (≈37 °C) XF DMEM media (Seahorse Bioscience #102353–100) supplemented with 2 mM L-Glutamine (BioNordika #BE17-605E) and pH adjusted to 7.4. The final concentration for each reagent used in this assay was 11 mM Glucose**,** 1.3 µM Oligomycin A, 0.1 M 2-Deoxy-Glucose (2-DG; Sigma-Aldrich #D8375-1G; *TeSlaa and Teitell, 2014*). Each measurement cycle consisted of a mixing time of 3 min and data acquisition period of 3 min (four data points). ECAR results refer to the average rates during each measurement cycle. Five baseline measurements were performed before adding each compound, and four response measurements were taken after each compound was added. Data was normalised to protein concentration using Seahorse Analytics software available at https://seahorseanalytics.agilent.com/ and the glycolysis rate was calculated by subtracting the ECAR value after 2-DG treatment from the ECAR value after addition of glucose.

## Purification of crude mitochondrial fraction

To purify crude mitochondrial fraction, cells expressing EGFP (control), mScarlet-RAB27A, SYTL5-EGFP and SYTL5(ΔC2AB)-EGFP were harvested from one 150 mm (80–90% confluency) dish, washed twice with ice-cold homogenisation buffer (210 mM mannitol (Sigma Aldrich #M4125), 70 mM sucrose

(Sigma Aldrich #S0389), 5 mM HEPES pH 7.12 at 25 °C) and harvested in 1 ml homogenisation buffer supplemented (HBS) with 1 mM EGTA (Sigma Aldrich #E3889) and 1 x protease inhibitor. The obtained cell lysate was further mechanically homogenised at 4 °C using a cell homogeniser (Isobiotech) equipped with a 16 µm clearance ball by repeatedly passing the cell lysate for 10 x repeats with the aid of two disposable syringes. The homogenised solution was further spun down at 1500 x *g* for 3 min at 4 °C to collect the TCL, and the obtained supernatant was collected and spun down at 13,000 x *g* for 20 min. The obtained supernatant corresponds to the cytosolic fraction, and the pellet was resuspended in 2 ml HBS and spun down at 13,000 x *g* for 20 min. The obtained pellet corresponds to the enriched mitochondrial fraction and was resuspended in 900 µl HBS. 50 µl of each cellular fraction was resuspended in 30 µl 2 x SDS-sample buffer and boiled at 100 °C for 5 min and further analysed using SDS-PAGE.

## GFP-TRAP and FLAG immunoprecipitation

Cells expressing EGFP were harvested from a 100 mm dish (80–90% confluency), washed with ice-cold PBS and scraped using ice-cold isolation buffer (1% NP-40, 2 mM EDTA, 136 mM NaCl, 20 mM Tris-HCl pH 7.4, 10% glycerol, 1 x protease inhibitor, and 1 x phosphatase inhibitor). Cell lysate was spun down at 20,000 x *g* for 10 min and the supernatant was added to 25 µl equilibrated GFP-trap agarose beads (Chromotek #gta-20). Samples were rotated end-over-end for 1 hr at 4 °C. After 4 x washing steps, the beads were collected by centrifugation at 2700 x *g* for 1 min, resuspended in 30 µl 2 x SDS-sample buffer and boiled at 100 °C for 5 min. The obtained supernatant was analysed by SDS-PAGE.

U2OS cells expressing pLVX-CMV-SYTL5-EGFP-3xFLAG, pLVX-CMV-SYTL5(ΔC2AB)-EGFP-3xFLAG and pLVX-CMV-SYTL5(ΔSHD)-EGFP-3xFLAG were harvested from a 150 mm dish (80–90% confluency), washed 2 x with cold PBS and scraped using 1 ml of cold lysis buffer 50 mM Tris-HCl, pH 7.4, 150 mM NaCl, 1% Triton X-100, 5 mM CaCl$_2$ Merck #1023821000, and 1 x protease inhibitor. Cell lysates were spun down at 20,000 x *g* for 10 min and the resulting supernatants were added to 40 µl of pre-washed anti-FLAG M2-Agarose Affinity Gel (Sigma-Aldrich #FLAGIPT-1). Samples were rotated end-over-end for 2 hr at 4 °C. After 3 x washing steps, the affinity gel was centrifuged at 5000 x *g* for 30 s and the FLAG fusion proteins eluted using 3 x FLAG peptide according to the manufacturer's protocol. The eluted FLAG fusion proteins were diluted in 4 ml 1 x TBST supplemented with 3% BSA and 5 mM CaCl$_2$.

## In vitro protein-lipid binding assay

Eluted FLAG fusion proteins SYTL5-EGFP-3xFLAG, SYTL5(ΔC2AB)-EGFP-3xFLAG and SYTL5(ΔSHD)-EGFP-3xFLAG were diluted in 1 x TBST supplemented with 3% BSA and 5 mM CaCl$_2$. PIP strips (Echelon Biosciences #P-6001) and membrane lipid strips (Echelon Biosciences #P-6002) were blocked in 1 x TBST supplemented with 3% BSA for 1 hr at room temperature and incubated with the diluted FLAG fusion protein for 1 hr at 4 °C with gentle agitation. Lipid membranes were washed 3 x with TBST and incubated with FLAG Tag primary antibody for 1 hr at room temperature and detected using SuperSignal West Dura Extended Duration Substrate (Thermo Fisher Scientific #34076).

## LC-MS/MS analysis of SYTL5-EGFP interactome

Cells expressing SYTL5-EGFP or EGFP (control) were harvested using NP-40 lysis buffer 10 mM Tris Cl pH 7.5, 150 mM NaCl, 0.5 mM EDTA, 0.5% NP-40, 1 x protease inhibitor. SYTL5-EGFP was immunoprecipitated using GFP-Trap agarose beads according to the manufacturer's protocol. The obtained beads were washed twice with 50 mM ammonium bicarbonate, reduced, alkylated and further digested by trypsin overnight at 37 °C. Digested peptides were transferred to a new tube, acidified and the peptides were de-salted for MS analysis.

LC-MS/MS analysis was carried out using a nanoElute nanoflow ultrahigh pressure LC system (Bruker Daltonics, Bremen, Germany) coupled to the timsTOF fleX mass spectrometer (Bruker Daltonics), using a CaptiveSpray nanoelectrospray ion source (Bruker Daltonics). De-salted peptides were loaded on a capillary C18 column Bruker (15 cm length, 75 µm inner diameter, 1.9 µm particle size, Bruker Daltonics). Peptides were separated at 50 °C using a 30 min gradient at a flow rate of 300 nl/min. The timsTOF fleX was operated in PASEF mode. Mass spectra for MS and MS/MS scans were recorded between m/z 100 and 1700. Ion mobility resolution was set to 0.60–1.60 V·s/cm over a ramp time of

100ms. Data-dependent acquisition was performed using 10 PASEF MS/MS scans per cycle with a near 100% duty cycle. A polygon filter was applied in the m/z and ion mobility space to exclude low m/z, singly charged ions from PASEF precursor selection. An active exclusion time of 0.4 min was applied to precursors that reached 20 000 intensity units. Collisional energy was ramped stepwise as a function of ion mobility.

Raw files from LC-MS/MS analyses were submitted to MaxQuant 1.6.17.0 software for peptide/protein identification. Parameters were set as follows: Carbamidomethyl (C) was set as a fixed modification, N-acetylation and methionine oxidation as variable modifications. First, search the error window of 20 ppm and mains search error of 6 ppm. Trypsin without proline restriction cleavage option was used, with two allowed miscleavages. Minimal unique peptides were set to one, and FDR allowed was 0.01 (1%) for peptide and protein identification. The UniProt human database was used. Generation of reversed sequences was selected to assign FDR rates. Further analysis was performed with Perseus (*Tyanova et al., 2016*), limma/voom (*Ritchie et al., 2015*) and Package R (*Dessau and Pipper, 2008*). Volcano plots were plotted with EnhancedVolcano (*Kevin Blighe, 2021*). GO analysis was performed with shinyGO (*Ge et al., 2020*). The mass spectrometry proteomics data have been deposited to the ProteomeXchange Consortium via the PRIDE (*Perez-Riverol et al., 2025*) partner repository with the dataset identifier PXD069382.

## Bulk gene expression analysis - normal and tumour tissue

Bulk gene expression distribution in healthy/normal tissue samples was collected from GTEx v8 (PMID: 26484571). We utilised a combined RNA-seq expression dataset of TCGA and GTEx samples available through the Xena platform to investigate SYTL5 and RAB27A expression in normal and tumour tissue (PMID: 32444850). Here, data from a total of n=77 samples provided normal adrenal gland expression (GTEx), and a total of n=128 samples provided expression in adrenocortical carcinoma samples (TCGA-ACC). Survival data from samples in the TCGA-ACC cohort (one record per patient) was obtained using the TCGABiolinks R package (*Colaprico et al., 2016*). Survival curve analysis was performed using GraphPad Prism 8.0.1 and the Logrank test (Mantel-Cox).

## Cortisol assay

To measure cortisol in cell culture supernates, siRNA knockdown by reverse transfection with siSYTL5 and siRAB27A was conducted as described earlier in NCI-H295R cells. Cells were harvested with TNTE Lysis Buffer. Cortisol from lysates was measured using the R&D Systems Cortisol Immunoassay kit (# KGE008B) as described in the protocol provided from the company. Controls were performed using cells treated with 50 µM Forskolin as a positive control to stimulate cortisol secretion (*Maglich et al., 2014*), or 10 µM mitotane to inhibit cortisol production (*Lin et al., 2012*).

## Statistical analysis

Statistical analysis was performed with GraphPad Prism (8.0.1, 9.3.1, and 10.2.0 versions) and the tests used are indicated in the respective figure legends. All values come from distinct samples.

****=$p < 0.0001$, ***=$p < 0.001$, **=$p < 0.01$, *=$p < 0.05$.

## Acknowledgements

We thank Jenni Laine and the Electron Microscopy Laboratory, Institute of Biomedicine, University of Turku, for assisting with EM sample preparation and TEM microscopy, Sachin Singh and Tuula Nyman at the OUS Proteomics Core Facility for assisting with mass spectrometry-based proteomic experiments, Ankush Sharma for help with analysis of mass spectrometry data, Anna Lång at ALM Core Facility Gaustad for assisting with widefield microscopy experiments, Santosh Phuyal for experimental assistance and Serhiy Pankiv for providing the lentiviral mScarlet-RAB expression constructs. We also thank Patricia González-Rodríguez for providing us with the U2OS pSu9-HaloTag7-mGFP cells and Viola Nähse for her contributions and advice regarding CRISPR knock-in strategies. This project was funded by the Research Council of Norway through its Centres of Excellence funding scheme (project number 262652) and FRIPRO grant (project number 249753), the Norwegian Cancer Society (project number 190251), Marie Skłodowska-Curie ETN grant under the European Union's Horizon 2020 Research and Innovation Programme (Grant Agreement No 765912 DRIVE) and by the University of Oslo Scientia Fellow program through the MSC scheme - Co-funding of Regional, National and

International Programmes (COFUND). Mass spectrometry-based proteomic analyses were performed by the Proteomics Core Facility, Department of Immunology, University of Oslo/Oslo University Hospital, which is supported by the Core Facilities program of the South-Eastern Norway Regional Health Authority. This core facility is also a member of the National Network of Advanced Proteomics Infrastructure (NAPI), which is funded by the Research Council of Norway INFRASTRUKTUR-program (project number: 295910).

---

## Additional information

### Funding

| Funder | Grant reference number | Author |
|---|---|---|
| Kreftforeningen | 190251 | Laura Trachsel-Moncho<br>Samuel J Rodgers<br>Anne Simonsen |
| Norges Forskningsråd | 262652 | Sigve Nakken<br>Anne Simonsen |
| Norges Forskningsråd | 249753 | Lauren Sophie Johnson<br>Matthew YW Ng<br>Anne Simonsen |
| HORIZON EUROPE Marie Sklodowska-Curie Actions | 10.3030/765912 | Ana Lapão |

The funders had no role in study design, data collection and interpretation, or the decision to submit the work for publication.

### Author contributions

Ana Lapão, Conceptualization, Data curation, Formal analysis, Validation, Investigation, Visualization, Methodology, Writing – original draft, Project administration; Lauren Sophie Johnson, Data curation, Formal analysis, Validation, Investigation, Visualization, Methodology, Writing – original draft, Project administration, Writing – review and editing; Laura Trachsel-Moncho, Matthew YW Ng, Eeva-Liisa Eskelinen, Investigation; Samuel J Rodgers, Methodology; Sakshi Singh, Sigve Nakken, Data curation, Formal analysis; Anne Simonsen, Conceptualization, Supervision, Funding acquisition, Writing – original draft, Project administration, Writing – review and editing

### Author ORCIDs

Ana Lapão https://orcid.org/0000-0002-5816-1240
Lauren Sophie Johnson https://orcid.org/0009-0007-8393-3154
Laura Trachsel-Moncho https://orcid.org/0000-0002-8525-7790
Samuel J Rodgers https://orcid.org/0000-0002-7186-8355
Sigve Nakken https://orcid.org/0000-0001-8468-2050
Eeva-Liisa Eskelinen https://orcid.org/0000-0003-0006-7785
Anne Simonsen https://orcid.org/0000-0003-4711-7057

Reviewer #1 (Public review): https://doi.org/10.7554/eLife.105541.3.sa1
Reviewer #2 (Public review): https://doi.org/10.7554/eLife.105541.3.sa2
Reviewer #3 (Public review): https://doi.org/10.7554/eLife.105541.3.sa3
Author response https://doi.org/10.7554/eLife.105541.3.sa4

---

## Additional files

### Supplementary files

MDAR checklist

Supplementary file 1. SYTL5-EGFP interactome.

## Data availability

All data generated or analysed during this study are included in the manuscript and supporting files; source data files have been provided for all figures. The mass spectrometry proteomics data have been deposited to the ProteomeXchange Consortium via the PRIDE partner repository with the dataset identifier PXD069382.

The following dataset was generated:

| Author(s) | Year | Dataset title | Dataset URL | Database and Identifier |
|---|---|---|---|---|
| Singh S | 2025 | The RAB27A effector SYTL5 regulates mitophagy and mitochondrial metabolism | https://www.ebi.ac.uk/pride/archive/projects/PXD069382 | PRIDE, PXD069382 |

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
