## [Editor Report · eLife Assessment]

This study by Lapao et al. uncovers a novel role for the Rab27A effector SYTL5 in regulating mitochondrial function and mitophagy under hypoxic conditions. Using a range of imaging and functional assays, the authors demonstrate that SYTL5 localizes to mitochondria in a Rab27A-dependent manner and impacts mitochondrial respiration and metabolic reprogramming. While the findings are **solid** and **valuable** in the area of cancer biology, further mechanistic clarity and improved imaging would strengthen the conclusions.

---

## [Referee Report · Reviewer #1 (Public review)]

Summary:

In this study, Ana Lapao et al. investigated the roles of Rab27 effector SYTL5 in cellular membrane trafficking pathways. The authors found that SYTL5 localizes to mitochondria in a Rab27A-dependent manner. They demonstrated that SYTL5-Rab27A positive vesicles containing mitochondrial material are formed under hypoxic conditions, thus they speculate that SYTL5 and Rab27A play roles in mitophagy. They also found that both SYTL5 and Rab27A are important for normal mitochondrial respiration. Cells lacking SYTL5 undergo a shift from mitochondrial oxygen consumption to glycolysis which is a common process known as the Warburg effect in cancer cells. Based on cancer patient database, the author noticed that low SYTL5 expression is related to reduced survival for adrenocortical carcinoma patients, indicating SYTL5 could be a negative regulator of the Warburg effect and potentially tumorigenesis.

Strengths:

The authors take advantages of multiple techniques and novel methods to perform the experiments.

(1) Live-cell imaging revealed that stably inducible expression of SYTL5 co-localized with filamentous structures positive for mitochondria. This result was further confirmed by using correlative light and EM (CLEM) analysis and western blotting from purified mitochondrial fraction.

(2) In order to investigate whether SYTL5 and RAB27A are required for mitophagy in hypoxic conditions, two established mitophagy reporter U2OS cell lines were used to analyze the autophagic flux.

Weaknesses:

This study revealed a potential function of SYTL5 in mitophagy and mitochondrial metabolism. However, the mechanistic evidence that establishes the relationship between SYTL5/Rab27A and mitophagy is insufficient. The involvement of SYTL5 in ACC needs more investigation. Furthermore, images and results supporting the major conclusions need to be improved.

Comments on revisions: The authors did not revise the paper as suggested.

---

## [Referee Report · Reviewer #2 (Public review)]

Summary:

The authors provide convincing evidence that Rab27 and STYL5 work together to regulate mitochondrial activity and homeostasis.

Strengths:

The development of models which allow the function to be dissected, and the rigous approach and testing of mitochondrial activity.

This work is carefully done, and supports the importance of the roles of Rab27A and STYL5.

---

## [Referee Report · Reviewer #3 (Public review)]

In the manuscript by Lapao et al., the authors uncover a role for the RAB27A effector protein SYTL5 in regulating mitochondrial function and apparent selective turnover of mitochondrial components. The authors find that SYTL5 localizes to mitochondria in a RAB27A dependent way and that loss of SYTL5 (or RAB27A) impairs lysosomal turnover of MTCO1 (but not a matrix-based reporter/other mitochondrial proteins). The authors go on to show that loss of SYTL5 impacts mitochondrial respiration and ECAR and as such may influence the Warburg effect and tumorigenesis. Of relevance here, the authors go on to show that SYTL5 expression is reduced in adrenocortical carcinomas and this correlates with reduced survival rates.

As previously reviewed, this is a very intriguing body of work and reveals a new role for SYTL5/RAB27A at the mitochondria. Unfortunately, it appears that SYTL5 is challenging protein to detect endogenously and the authors' cell lines "comprise a heterogenous pool with high variability", which means that a lot of my original concerns remain. It is still also not clear if the conventional autophagy machinery is required for this pathway, especially if SYTL5/RAB27A mitochondrial recruitment is upstream of this. Hopefully, in future work, the authors (and/or others) will be able to address this and build on the mechanisms of this interesting and potentially important pathway.

---

## [Author Response]

The following is the authors’ response to the original reviews.

**Reviewer #1 (Public review):**
Summary:In this study, Ana Lapao et al. investigated the roles of Rab27 effector SYTL5 in cellular membrane trafficking pathways. The authors found that SYTL5 localizes to mitochondria in a Rab27A-dependent manner. They demonstrated that SYTL5-Rab27A positive vesicles containing mitochondrial material are formed under hypoxic conditions, thus they speculate that SYTL5 and Rab27A play roles in mitophagy. They also found that both SYTL5 and Rab27A are important for normal mitochondrial respiration. Cells lacking SYTL5 undergo a shift from mitochondrial oxygen consumption to glycolysis which is a common process known as the Warburg effect in cancer cells. Based on the cancer patient database, the author noticed that low SYTL5 expression is related to reduced survival for adrenocortical carcinoma patients, indicating SYTL5 could be a negative regulator of the Warburg effect and potentially tumorigenesis.Strengths:The authors take advantage of multiple techniques and novel methods to perform the experiments.(1) Live-cell imaging revealed that stably inducible expression of SYTL5 co-localized with filamentous structures positive for mitochondria. This result was further confirmed by using correlative light and EM (CLEM) analysis and western blotting from purified mitochondrial fraction.(2) In order to investigate whether SYTL5 and Rab27A are required for mitophagy in hypoxic conditions, two established mitophagy reporter U2OS cell lines were used to analyze the autophagic flux.Weaknesses:This study revealed a potential function of SYTL5 in mitophagy and mitochondrial metabolism. However, the mechanistic evidence that establishes the relationship between SYTL5/Rab27A and mitophagy is insufficient. The involvement of SYTL5 in ACC needs more investigation. Furthermore, images and results supporting the major conclusions need to be improved.

We thank the reviewer for their constructive comments. We agree that a complete understanding of the mechanism by which SYTL5 and Rab27A are recruited to the mitochondria and subsequently involved in mitophagy requires further investigation. Here, we have shown that SYTL5 recruitment to the mitochondria requires both its lipid-binding C2 domains and the Rab27A-binding SHD domain (Figure 1G-H). This implies a coincidence detection mechanism for mitochondrial localisation of SYTL5. Additionally, we find that mitochondrial recruitment of SYTL5 is dependent on the GTPase activity and mitochondrial localisation of Rab27A (Figure 2D-E). We also identified proteins linked to the cellular response to oxidative stress, reactive oxygen species metabolic process, regulation of mitochondrion organisation and protein insertion into mitochondrial membrane to be enriched in the SYTL5 interactome (Figure 3A and C).

However, less details regarding the mitochondrial localisation of Rab27A are understood. To investigate this, we have now performed a mass spectrometry analysis to identify the interactome of Rab27A (see Author response table 1 below,). U2OS cells with stable expression of mScarlet-Rab27A or mScarlet only, were subjected to immunoprecipitation, followed by MS analysis. Of the 32 significant Rab27A-interacting hits (compared to control), two of the hits are located in the inner mitochondrial membrane (IMM); ATP synthase F(1) complex subunit alpha (P25705), and mitochondrial very long-chain specific acyl-CoA dehydrogenase (VLCAD)(P49748). However, as these IMM proteins are not likely involved in mitochondrial recruitment of Rab27A, observed under basal conditions, we choose not to include these data in the manuscript.

It is known that other RAB proteins are recruited to the mitochondria. During parkin-mediated mitophagy, RABGEF1 (a guanine nucleotide exchange factor) is recruited through its ubiquitin-binding domain and directs mitochondrial localisation of RAB5, which subsequently leads to recruitment of RAB7 by the MON1/CCZ1 complex[1]. As already mentioned in the discussion (p. 12), ubiquitination of the Rab27A GTPase activating protein alpha (TBC1D10A) is reduced in the brain of Parkin KO mouse compared to controls[35], suggesting a possible connection of Rab27A with regulatory mechanisms that are linked with mitochondrial damage and dysfunction. While this an interesting avenue to explore, in this paper we will not follow up further on the mechanism of mitochondrial recruitment of Rab27A.

**Author response table 1. sa4table1:** Rab27A interactome. Proteins co-immunoprecipitated with mScarlet-Rab27A vs mScarlet expressing control. The data show average of three replicates.

Gene ID	log FC	PValue	Protein ID
SYTL1	6.51279398	2.58011E-07	Q8IYJ3
SYTL4	9.462240971	5.37442E-07	Q96C24
RAB27A	9.964034476	0.000225708	P51159
PHKA1	1.901649808	0.00024689	P46020
SYTL5	3.382670586	0.00028503	Q8TDW5
RPN1	1.842817754	0.000618174	P04843
SYTL2	2.787103426	0.000976248	Q9HCH5
SYTL3	1.763916791	0.001231698	Q4VX76
YWHAB	1.324625725	0.001792747	P31946
KPNA4	-2.119041679	0.001819937	000629
KPNA2	-1.316606494	0.002674273	P52292
XPO7	1.881758901	0.00434815	Q9UIA9
H3F3C	-1.014289683	0.005676851	Q6NXT2
FTSJ3	-1.536695759	0.013027871	Q8IY81
HSPA5	-0.696530058	0.013830063	P11021
GCN1L1	1.282942068	0.014007173	Q92616
SEC23A	-1.154781691	0.014551903	Q15436
YWHAE	1.051069572	0.014721005	P62258
CSE1L	1.573274711	0.015215159	P55060
PRKDC	1.203932332	0.015522477	P78527
HSD17B4	0.622782422	0.018684219	P51659
TUBA1C	0.757418802	0.021056742	Q9BQE3
ATP5A1	1.652450437	0.021124762	P25705
ACADVL	1.487591887	0.023527244	P49748
SYNE1	1.484495661	0.023619887	Q8NF91
TUBA1B	0.869686638	0.026093033	P68363
KHNYN	0.530428593	0.032980211	015037
STIP1	-1.003028534	0.037684326	P31948
RNF213	2.063787445	0.041437396	Q63HN8
MACF1	-0.558906019	0.041690413	Q9UPN3
BANF1	-2.961226325	0.047019044	075531
AGL	-1.021197131	0.047045989	P35573

To investigate the role of SYTL5 in the context of ACC, we acquired the NCI-H295R cell line isolated from the adrenal gland of an adrenal cancer patient. The cells were cultured as recommended from ATCC using DMEM/F-12 supplemented with NuSerum and ITS +premix. It is important to note that the H295R cells were adapted to grow as an adherent monolayer from the H295 cell line which grows in suspension. However, there can still be many viable H295R cells in the media.

We attempted to conduct OCR and ECAR measurements using the Seahorse XF upon knockdown of SYTL5 and/or Rab27A in H295R cells. For these assays, it is essential that the cells be seeded in a monolayer at 70-90% confluency with no cell clusters[4]. Poor adhesion of the cells can cause inaccurate measurements by the analyser. Unfortunately, the results between the five replicates we carried out were highly inconsistent, the same knockdown produced trends in opposite directions in different replicates. This is likely due to problems with seeding the cells. Despite our best efforts to optimise seeding number, and pre-coating the plate with poly-D-lysine[5] we observed poor attachment of cells and inability to form a monolayer.

To study the localisation of SYTL5 and Rab27A in an ACC model, we transduced the H295R cells with lentiviral particles to overexpress pLVX-SV40-mScarlet-I-Rab27A and pLVX-CMV-SYTL5-EGFP-3xFLAG. Again, this proved unsuccessful after numerous attempts at optimising transduction.

These issues limited our investigation into the role of SYTL5 in ACC to the cortisol assay (Supplementary Figure 6). For this the H295R cells were an appropriate model as they are able to produce an array of adrenal cortex steroids[6] including cortisol[7]. In this assay, measurements are taken from cell culture supernatants, so the confluency of the cells does not prevent consistent results as the cortisol concentration was normalised to total protein per sample. With this assay we were able to rule out a role for SYTL5 and Rab27A in the secretion of cortisol.

Another consideration when investigating the involvement of SYTL5 in ACC, is that in general ACC cells should have a low expression of SYTL5 as is seen from the patient expression data (Figure 6B).

The reviewer also writes “Furthermore, images and results supporting the major conclusions need to be improved.”. We have tried several times, without success, to generate U2OS cells with CRISPR/Cas9-mediated C-terminal tagging of endogenous SYTL5 with mNeonGreen, using an approach that has been successfully implemented in the lab for other genes. This is likely due to a lack of suitable sgRNAs targeting the C-terminal region of SYTL5, which have a low predicted efficiency score and a large number of predicted off-target sites in the human genome including several other gene exons and introns (see Author response image 2).

We have also included new data (Supplementary Figure 4B) showing that some of the hypoxia-induced SYTL5-Rab27A-positive vesicles stain positive for the autophagy markers p62 and LC3B when inhibiting lysosomal degradation, further strengthening our data that SYTL5 and Rab27A function as positive regulators of mitophagy.

**Reviewer #2 (Public review):**
Summary:The authors provide convincing evidence that Rab27 and STYL5 work together to regulate mitochondrial activity and homeostasis.Strengths:The development of models that allow the function to be dissected, and the rigorous approach and testing of mitochondrial activity.Weaknesses:There may be unknown redundancies in both pathways in which Rab27 and SYTL5 are working which could confound the interpretation of the results.Suggestions for revision:Given that Rab27A and SYTL5 are members of protein families it would be important to exclude any possible functional redundancies coming from Rab27B expression or one of the other SYTL family members. For Rab27 this would be straightforward to test in the assays shown in Figure 4 and Supplementary Figure 5. For SYTL5 it might be sufficient to include some discussion about this possibility.

We thank the reviewer for pointing out the potential redundancy issue for Rab27A and SYTL5. There are multiple studies demonstrating the redundancy between Rab27A and Rab27B. For example, in a study of the disease Griscelli syndrome, caused by Rab27A loss of function, expression of either Rab27A or Rab27B rescues the healthy phenotype indicating redundancy[8]. This redundancy however applies to certain function and cell types. In fact, in a study regarding hair growth, knockdown of Rab27B had the opposite effect to knockdown of Rab27A[9].

In this paper, we conducted all assays in U2OS cells, in which the expression of Rab27B is very low. Human Protein Atlas reports expression of 0.5nTPM for Rab27B, compared to 18.4nTPM for Rab27A. We also observed this low level of expression of Rab27B compared to Rab27A by qPCR in U2OS cells. Therefore, there would be very little endogenous Rab27B expression in cells depleted of Rab27A (with siRNA or KO). In line with this, Rab27B peptides were not detected in our SYTL5 interactome MS data (Table 1 in paper). Moreover, as Rab27A depletion inhibits mitochondrial recruitment of SYTL5 and mitophagy, it is not likely that Rab27B provides a functional redundancy. It is possible that Rab27B overexpression could rescue mitochondrial localisation of SYTL5 in Rab27A KO cells, but this was not tested as we do not have any evidence for a role of Rab27B in these cells. Taken together, we believe our data imply that Rab27B is very unlikely to provide any functional redundancy to Rab27A in our experiments.

For the SYTL family, all five members are Rab27 effectors, binding to Rab27 through their SHD domain. Together with Rab27, all SYTL’s have been implicated in exocytosis in different cell types. For example, SYTL1 in exocytosis of azurophilic granules from neutrophils[10], SYTL2 in secretion of glucagon granules from pancreatic α cells[11], SYTL3 in secretion of lytic granules from cytotoxic T lymphocytes[12], SYTL4 in exocytosis of dense hormone containing granules from endocrine cells[13] and SYTL5 in secretion of the RANKL cytokine from osteoblasts[14]. This indicates a potential for redundancy through their binding to Rab27 and function in vesicle secretion/trafficking. However, one study found that different Rab27 effectors have distinct functions at different stages of exocytosis[15].

Very little known about redundancy or hierarchy between these proteins. Differences in function may be due to the variation in gene expression profile across tissues for the different SYTL’s (see Author response image 1 below). SYTL5 is enriched in the brain unlike the others, suggesting possible tissue specific functions. There are also differences in the binding affinities and calcium sensitivities of the C2iA and C2B domains between the SYTL proteins[16].

**Author response image 1. sa4fig1:** GTEx Multi Gene Query for SYTL1-5.

All five SYTL’s are expressed in the U2OS cell line with nTPMs according to Human Protein Atlas of SYTL1: 7.5, SYTL2: 13.4, SYTL3:14.2, SYTL4: 8.7, SYTL5: 4.8. In line with this, in the Rab27A interactome, when comparing cells overexpressing mScarlet-Rab27A with control cells, we detected all five SYTL’s as specific Rab27A-interacting proteins (see Author response table 1 above). Whereas, in the SYTL5 interactome we did not detect any other SYTL protein (table 1 in paper), confirming that they do not form a complex with SYTL5.

We have included the following text in the discussion (p. 12): “SYTL5 and Rab27A are both members of protein families, suggesting possible functional redundancies from Rab27B or one of the other SYTL isoforms. While Rab27B has a very low expression in U2OS cells, all five SYTL’s are expressed. However, when knocking out or knocking down SYTL5 and Rab27A we observe significant effects that we presume would be negated if their isoforms were providing functional redundancies. Moreover, we did not detect any other SYTL protein or Rab27B in the SYTL5 interactome, confirming that they do not form a complex with SYTL5.”

Suggestions for Discussion:Both Rab27A and STYL5 localize to other membranes, including the endolysosomal compartments. How do the authors envisage the mechanism or cellular modifications that allow these proteins, either individually or in complex to function also to regulate mitochondrial funcYon? It would be interesYng to have some views.

We agree that it would be interesting to better understand the mechanism involved in modulation of the localisation and function of SYTL5 and Rab27A at different cellular compartments, including the mitochondria. Here, we have shown that SYTL5 recruitment to the mitochondria involves coincidence detection, as both its lipid-binding C2 domains and the Rab27A-binding SHD domain are required (Figure 1G-H). Both these domains also seem required for localisation of SYTL5 to vesicles, and we can only speculate that binding to different lipids (Figure 1F) may regulate SYTL5 localisation. Additionally, we find that mitochondrial recruitment of SYTL5 is dependent on the GTPase activity and mitochondrial localisation of Rab27A (Figure 2D-E). However, this seems also the case for vesicular recruitment of SYTL5, although a few SYTL5-Rab27A (T23N) positive vesicles were seen (Figure 2E).

To characterise the mechanisms involved in mitochondrial localisation of Rab27A, we have performed mass spectrometry analysis to identify the interactome of Rab27A (see Author response table 1 above). U2OS cells with stable expression of mScarlet-Rab27A or mScarlet only were subjected to immunoprecipitation, followed by MS analysis. Of the 32 significant Rab27A-interacting hits (compared to control), two of the hits localise in the inner mitochondrial membrane (IMM); ATP synthase F(1) complex subunit alpha (P25705), and mitochondrial very long-chain specific acyl-CoA dehydrogenase (VLCAD)(P49748). However, as these IMM proteins are not likely involved in mitochondrial recruitment of Rab27A, observed under basal conditions, we chose not to include these data in the manuscript.

It is known that other RAB proteins are recruited to the mitochondria by regulation of their GTPase activity. During parkin-mediated mitophagy, RABGEF1 (a guanine nucleotide exchange factor) is recruited through its ubiquitin-binding domain and directs mitochondrial localisation of RAB5, which subsequently leads to recruitment of RAB7 by the MON1/CCZ1 GEF complex[1]. As already mentioned in the discussion (p.12), ubiquitination of the Rab27A GTPase activating protein alpha (TBC1D10A) is reduced in the brain of Parkin KO mouse compared to controls[35], suggesting a possible connection of Rab27A with regulatory mechanisms that are linked with mitochondrial damage and dysfunction. While this an interesting avenue to explore, it is beyond the scope of this paper.

Our data suggest that SYTL5 functions as a negative regulator of the Warburg effect, the switch from OXPHOS to glycolysis. While both SYTL5 and Rab27A seem required for mitophagy of selective mitochondrial components, and their depletion leading to reduced mitochondrial respiration and ATP production, only depletion of SYTL5 caused a switch to glycolysis. The mechanisms involved are unclear, but we found several proteins linked to the cellular response to oxidative stress, reactive oxygen species metabolic process, regulation of mitochondrion organisation and protein insertion into mitochondrial membrane to be enriched in the SYTL5 interactome (Figure 3A and C).

We have addressed this comment in the discussion on p.12

**Reviewer #3 (Public review):**
Summary:In the manuscript by Lapao et al., the authors uncover a role for the Rab27A effector protein SYTL5 in regulating mitochondrial function and turnover. The authors find that SYTL5 localizes to mitochondria in a Rab27A-dependent way and that loss of SYTL5 (or Rab27A) impairs lysosomal turnover of an inner mitochondrial membrane mitophagy reporter but not a matrix-based one. As the authors see no co-localization of GFP/mScarlet tagged versions of SYTL5 or Rab27A with LC3 or p62, they propose that lysosomal turnover is independent of the conventional autophagy machinery. Finally, the authors go on to show that loss of SYTL5 impacts mitochondrial respiration and ECAR and as such may influence the Warburg effect and tumorigenesis. Of relevance here, the authors go on to show that SYTL5 expression is reduced in adrenocortical carcinomas and this correlates with reduced survival rates.Strengths:There are clearly interesting and new findings here that will be relevant to those following mitochondrial function, the endocytic pathway, and cancer metabolism.Weaknesses:The data feel somewhat preliminary in that the conclusions rely on exogenously expressed proteins and reporters, which do not always align.As the authors note there are no commercially available antibodies that recognize endogenous SYTL5, hence they have had to stably express GFP-tagged versions. However, it appears that the level of expression dictates co-localization from the examples the authors give (though it is hard to tell as there is a lack of any kind of quantitation for all the fluorescent figures). Therefore, the authors may wish to generate an antibody themselves or tag the endogenous protein using CRISPR.

We agree that the level of SYTL5 expression is likely to affect its localisation. As suggested by the reviewer, we have tried hard, without success, to generated U2OS cells with CRISPR knock-in of a mNeonGreen tag at the C-terminus of endogenous SYTL5, using an approach that has been successfully implemented in the lab for other genes. This is likely due to a lack of suitable sgRNAs targeting the C-terminal region of SYTL5, which have a low predicted efficiency score and a large number of predicted off-target sites in the human genome including several other gene exons and introns (see Author response image 2).

**Author response image 2. sa4fig2:** Overview of sgRNAs targeting the C-terminal region of SYTL5.

Although the SYTL5 expression level might affect its cellular localization, we also found the mitochondrial localisation of SYTL5-EGFP to be strongly increased in cells co-expressing mScarletRab27A, supporting our findings of Rab27A-mediated mitochondrial recruitment of SYTL5. We have also included new data (Supplementary Figure 4B) showing that some of the hypoxia-induced SYTL5Rab27A-positive vesicles stain positive for the autophagy markers p62 and LC3B when inhibiting lysosomal degradation, further strengthening our data that SYTL5 and Rab27A function as positive regulators of mitophagy.

In relation to quantitation, the authors found that SYTL5 localizes to multiple compartments or potentially a few compartments that are positive for multiple markers. Some quantitation here would be very useful as it might inform on function.

We find that SYTL5-EGFP localizes to mitochondria, lysosomes and the plasma membrane in U2OS cells with stable expression of SYTL5-EGFP and in SYTL5/Rab27A double knock-out cells rescued with SYTL5EGFP and mScralet-Rab27A. We also see colocalization of SYTL5-EGFP with endogenous p62, LC3 and LAMP1 upon induction of mitophagy. However, as these cell lines comprise a heterogenous pool with high variability we do not believe that quantification of the overexpressing cell lines would provide beneficial information in this scenario. As described above, we have tried several times to generate SYTL5 knock-in cells without success.

The authors find that upon hypoxia/hypoxia-like conditions that punctate structures of SYTL5 and Rab27A form that are positive for Mitotracker, and that a very specific mitophagy assay based on pSu9-Halo system is impaired by siRNA of SYTL5/Rab27A, but another, distinct mitophagy assay (Matrix EGFP-mCherry) shows no change. I think this work would strongly benefit from some measurements with endogenous mitochondrial proteins, both via immunofluorescence and western blot-based flux assays.

In addition to the western blotting for different endogenous ETC proteins showing significantly increased levels of MTCO1 in cells depleted of SYTL5 and/or Rab27A (Figure 5E-F), we have now blotted for the endogenous mitochondrial proteins, COXIV and BNIP3L, in DFP and DMOG conditions upon knockdown of SYTL5 and/or Rab27A (Figure 5G and Supplementary Figure 5A). Although there was a trend towards increased levels, we did not see any significant changes in total COXIV or BNIP3L levels when SYTL5, Rab27A or both are knocked down compared to siControl. Blotting for endogenous mitochondrial proteins is however not the optimum readout for mitophagy. A change in mitochondrial protein level does not necessarily result from mitophagy, as other factors such as mitochondrial biogenesis and changes in translation can also have an effect. Mitophagy is a dynamic process, which is why we utilise assays such as the HaloTag and mCherry-EGFP double tag as these indicate flux in the pathway. Additionally, as mitochondrial proteins have different half-lives, with many long-lived mitochondrial proteins[17], differences in turnover rates of endogenous proteins make the results more difficult to interpret.

A really interesting aspect is the apparent independence of this mitophagy pathway on the conventional autophagy machinery. However, this is only based on a lack of co-localization between p62or LC3 with LAMP1 and GFP/mScarlet tagged SYTL5/Rab27A. However, I would not expect them to greatly colocalize in lysosomes as both the p62 and LC3 will become rapidly degraded, while the eGFP and mScarlet tags are relatively resistant to lysosomal hydrolysis. -/+ a lysosome inhibitor might help here and ideally, the functional mitophagy assays should be repeated in autophagy KOs.

We thank the reviewer for this suggestion. We have now repeated the colocalisation studies in cells treated with DFP with the addition of bafilomycin A1 (BafA1) to inhibit the lysosomal V-ATPase. Indeed, we find that a few of the SYTL5/Rab27A/MitoTracker positive structures also stain positive for p62 and LC3 (Supplementary Figure 4B). As expected, the occurrence of these structures was rare, as BafA1 was only added for the last 4 hrs of the 24 hr DFP treatment. However, we cannot exclude the possibility that there are two different populations of these vesicles.

The link to tumorigenesis and cancer survival is very interesYng but it is not clear if this is due to the mitochondrially-related aspects of SYTL5 and Rab27A. For example, increased ECAR is seen in the SYTL5 KO cells but not in the Rab27A KO cells (Fig.5D), implying that mitochondrial localization of SYTL5 is not required for the ECAR effect. More work to strengthen the link between the two sections in the paper would help with future direcYons and impact with respect to future cancer treatment avenues to explore.

We agree that the role of SYTL5 in ACC requires future investigation. While we observe reduced OXPHOS levels in both SYTL5 and Rab27A KO cells (Figure 5B), glycolysis was only increased in SYTL5 KO cells (Figure 5D). We believe this indicates that Rab27A is being negatively regulated by SYTL5, as ECAR was unchanged in both the Rab27A KO and Rab27A/SYTL5 dKO cells. This suggests that Rab27A is required for the increase in ECAR when SYTL5 is depleted, therefore SYTL5 negatively regulates Rab27A. The mechanism involved is unclear, but we found several proteins linked to the cellular response to oxidative stress, reactive oxygen species metabolic process, regulation of mitochondrion organisation and protein insertion into mitochondrial membrane to be enriched in the SYTL5 interactome (Figure 3A and C).

To investigate the link to cancer further, we tested the effect of knockdown of SYTL5 and/or Rab27A on the levels of mitochondrial ROS. ROS levels were measured by flow cytometry using the MitoSOX Red dye, together with the MitoTracker Green dye to normalise ROS levels to the total mitochondria. Cells were treated with the antioxidant N-acetylcysteine (NAC)[18] as a negative control and menadione as a positive control, as menadione induces ROS production via redox cycling[19]. We must consider that there is also a lot of autofluorescence from cells that makes it impossible to get a level of ‘zero ROS’ in this experiment. We did not see a change in ROS with knockdown of SYTL5 and/or Rab27A compared to the NAC treated or siControl samples (see Author response image 3 below). The menadione samples confirm the success of the experiment as ROS accumulated in these cells. Thus, based on this, we do not believe that low SYTL5 expression would affect ROS levels in ACC tumours.

**Author response image 3. sa4fig3:** Mitochondrial ROS production normalised to total mitochondria.

As discussed in our response to Reviewer #1, we tried hard to characterise the role of SYTL5 in the context of ACC using the NCI-H295R cell line isolated from the adrenal gland of an adrenal cancer patient. We attempted to conduct OCR and ECAR measurements using the Seahorse XF upon knockdown of SYTL5 and/or Rab27A in H295R cells without success, due to poor attachment of the cells and inability to form a monolayer. We also transduced the H295R cells with lentiviral particles to overexpress pLVX-SV40-mScarlet-I-Rab27A and pLVX-CMV-SYTL5-EGFP-3xFLAG to study the localisation of SYTL5 and Rab27A in an ACC model. Again, this proved unsuccessful after numerous attempts at optimising the transduction. These issues limited our investigation into the role of SYTL5 in ACC to the cortisol assay (Supplementary Figure 6). For this the H295R cells were an appropriate model as they are able to produce an array of adrenal cortex steroids[6] including cortisol[7] In this assay, measurements are taken from cell culture supernatants, so the confluency of the cells does not prevent consistent results as the cortisol concentration was normalised to total protein per sample. With this assay we were able to rule out a role for SYTL5 and Rab27A in the secretion of cortisol.

Another consideration when investigating the involvement of SYTL5 in ACC, is that in general ACC cells should have a low expression of SYTL5 as is seen from the patient expression data (Figure 6B).

Further studies into the link between SYTL5/Rab27A and cancer are beyond the scope of this paper as we are limited to the tools and expertise available in the lab.

References

(1) Yamano, K. et al. Endosomal Rab cycles regulate Parkin-mediated mitophagy. eLife 7 (2018). https://doi.org:10.7554/eLife.31326

(2) Carré, M. et al. Tubulin is an inherent component of mitochondrial membranes that interacts with the voltage-dependent anion channel. The Journal of biological chemistry 277, 33664-33669 (2002). https://doi.org:10.1074/jbc.M203834200

(3) Hoogerheide, D. P. et al. Structural features and lipid binding domain of tubulin on biomimetic mitochondrial membranes. Proceedings of the National Academy of Sciences 114, E3622-E3631 (2017). https://doi.org:10.1073/pnas.1619806114

(4) Plitzko, B. & Loesgen, S. Measurement of Oxygen Consumption Rate (OCR) and Extracellular Acidification Rate (ECAR) in Culture Cells for Assessment of the Energy Metabolism. Bio Protoc 8, e2850 (2018). https://doi.org:10.21769/BioProtoc2850

(5) Yavin, E. & Yavin, Z. Attachment and culture of dissociated cells from rat embryo cerebral hemispheres on polylysine-coated surface. The Journal of cell biology 62, 540-546 (1974). https://doi.org:10.1083/jcb.62.2.540

(6) Wang, T. & Rainey, W. E. Human adrenocortical carcinoma cell lines. Mol Cell Endocrinol 351, 5865 (2012). https://doi.org:10.1016/j.mce.2011.08.041

(7) Rainey, W. E. et al. Regulation of human adrenal carcinoma cell (NCI-H295) production of C19 steroids. J Clin Endocrinol Metab 77, 731-737 (1993). https://doi.org:10.1210/jcem.77.3.8396576

(8) Barral, D. C. et al. Functional redundancy of Rab27 proteins and the pathogenesis of Griscelli syndrome. J. Clin. Invest. 110, 247-257 (2002). https://doi.org:10.1172/jci15058

(9) Ku, K. E., Choi, N. & Sung, J. H. Inhibition of Rab27a and Rab27b Has Opposite Effects on the Regulation of Hair Cycle and Hair Growth. Int. J. Mol. Sci. 21 (2020). https://doi.org:10.3390/ijms21165672

(10) Johnson, J. L., Monfregola, J., Napolitano, G., Kiosses, W. B. & Catz, S. D. Vesicular trafficking through cortical actin during exocytosis is regulated by the Rab27a effector JFC1/Slp1 and the RhoA-GTPase–activating protein Gem-interacting protein. Mol. Biol. Cell 23, 1902-1916 (2012). https://doi.org:10.1091/mbc.e11-12-1001

(11) Yu, M. et al. Exophilin4/Slp2-a targets glucagon granules to the plasma membrane through unique Ca2+-inhibitory phospholipid-binding activity of the C2A domain. Mol. Biol. Cell 18, 688696 (2007). https://doi.org:10.1091/mbc.e06-10-0914

(12) Kurowska, M. et al. Terminal transport of lyXc granules to the immune synapse is mediated by the kinesin-1/Slp3/Rab27a complex. Blood 119, 3879-3889 (2012). https://doi.org:10.1182/blood-2011-09-382556

(13) Zhao, S., Torii, S., Yokota-Hashimoto, H., Takeuchi, T. & Izumi, T. Involvement of Rab27b in the regulated secretion of pituitary hormones. Endocrinology 143, 1817-1824 (2002). https://doi.org:10.1210/endo.143.5.8823

(14) Kariya, Y. et al. Rab27a and Rab27b are involved in stimulation-dependent RANKL release from secretory lysosomes in osteoblastic cells. J Bone Miner Res 26, 689-703 (2011). https://doi.org:10.1002/jbmr.268

(15) Zhao, K. et al. Functional hierarchy among different Rab27 effectors involved in secretory granule exocytosis. Elife 12 (2023). https://doi.org:10.7554/eLife.82821

(16) Izumi, T. Physiological roles of Rab27 effectors in regulated exocytosis. Endocr J 54, 649-657 (2007). https://doi.org:10.1507/endocrj.kr-78

(17) Bomba-Warczak, E. & Savas, J. N. Long-lived mitochondrial proteins and why they exist. Trends in cell biology 32, 646-654 (2022). https://doi.org:10.1016/j.tcb.2022.02.001

(18) Curtin, J. F., Donovan, M. & Cotter, T. G. Regulation and measurement of oxidative stress in apoptosis. Journal of Immunological Methods 265, 49-72 (2002). https://doi.org:https://doi.org/10.1016/S0022-1759(02)00070-4

(19) Criddle, D. N. et al. Menadione-induced Reative Oxygen Species Generation via Redox Cycling Promotes Apoptosis of Murine Pancreatic Acinar Cells. Journal of Biological Chemistry 281, 40485-40492 (2006). https://doi.org:https://doi.org/10.1074/jbc.M607704200